# CERTIFIED DEFENCES AGAINST ADVERSARIAL PATCH ATTACKS ON SEMANTIC SEGMENTATION

**Maksym Yatsura[1,2]\***, **Kaspar Sakmann[1]**, **N. Grace Hua[1]**, **Matthias Hein[2,3]**, **Jan Hendrik Metzen[1]**
[1]Bosch Center for Artificial Intelligence, Robert Bosch GmbH,
[2]University of Tübingen, [3]Tübingen AI Center

## ABSTRACT

Adversarial patch attacks are an emerging security threat for real world deep learning applications. We present DEMASKED SMOOTHING, the first approach (up to our knowledge) to certify the robustness of semantic segmentation models against this threat model. Previous work on certifiably defending against patch attacks has mostly focused on image classification task and often required changes in the model architecture and additional training which is undesirable and computationally expensive. In DEMASKED SMOOTHING, any segmentation model can be applied without particular training, fine-tuning, or restriction of the architecture. Using different masking strategies, DEMASKED SMOOTHING can be applied both for certified detection and certified recovery. In extensive experiments we show that DEMASKED SMOOTHING can on average certify 63% of the pixel predictions for a 1% patch in the detection task and 46% against a 0.5% patch for the recovery task on the ADE20K dataset.

## 1 INTRODUCTION

Physically realizable adversarial attacks are a threat for safety-critical (semi-)autonomous systems such as self-driving cars or robots. Adversarial patches (Brown et al., 2017; Karmon et al., 2018) are the most prominent example of such an attack. Their realizability has been demonstrated repeatedly, for instance by Lee & Kolter (2019): an attacker places a printed version of an adversarial patch in the physical world to fool a deep learning system. While empirical defenses (Hayes, 2018; Naseer et al., 2019; Selvaraju et al., 2019; Wu et al., 2020) may offer robustness against known attacks, it does not provide any guarantees against unknown future attacks (Chiang et al., 2020). Thus, certified defenses for the patch threat model, which allow guaranteed robustness against all possible attacks for the given threat model, are crucial for safety-critical applications.

Research on certifiable defenses against adversarial patches can be broadly categorized into certified recovery and certified detection. *Certified recovery* (Chiang et al., 2020; Levine & Feizi, 2020; Zhang et al., 2020; Xiang et al., 2021; Metzen & Yatsura, 2021; Lin et al., 2021; Xiang et al., 2022a; Salman et al., 2021; Chen et al., 2022) has the objective to make a correct prediction on an input even in the presence of an adversarial patch. In contrast, *certified detection* (McCoyd et al., 2020; Xiang & Mittal, 2021b; Han et al., 2021; Huang & Li, 2021) provides a weaker guarantee by only aiming at *detecting* inputs containing adversarial patches. While certified recovery is more desirable in principle, it typically comes at a high cost of reduced performance on clean data. In practice, certified detection might be preferable because it allows maintaining high clean performance. Most existing certifiable defenses against patches are focused on image classification, with the exception of DetectorGuard (Xiang & Mittal, 2021a) and ObjectSeeker (Xiang et al., 2022b) that certifiably defend against patch hiding attacks on object detectors. Moreover, existing defences are not easily applicable to arbitrary downstream models, because they assume either that the downstream model is trained explicitly for being certifiably robust (Levine & Feizi, 2020; Metzen & Yatsura, 2021), or that the model has a certain network architecture such as BagNet (Zhang et al., 2020; Metzen & Yatsura, 2021; Xiang et al., 2021) or a vision transformer (Salman et al., 2021; Huang & Li, 2021). A notable exception is PatchCleanser (Xiang et al., 2022a), which can be combined with arbitrary downstream models but is restricted to image classification.

---

\*Correspondence to: Maksym Yatsura <maksym.yatsura@de.bosch.com>

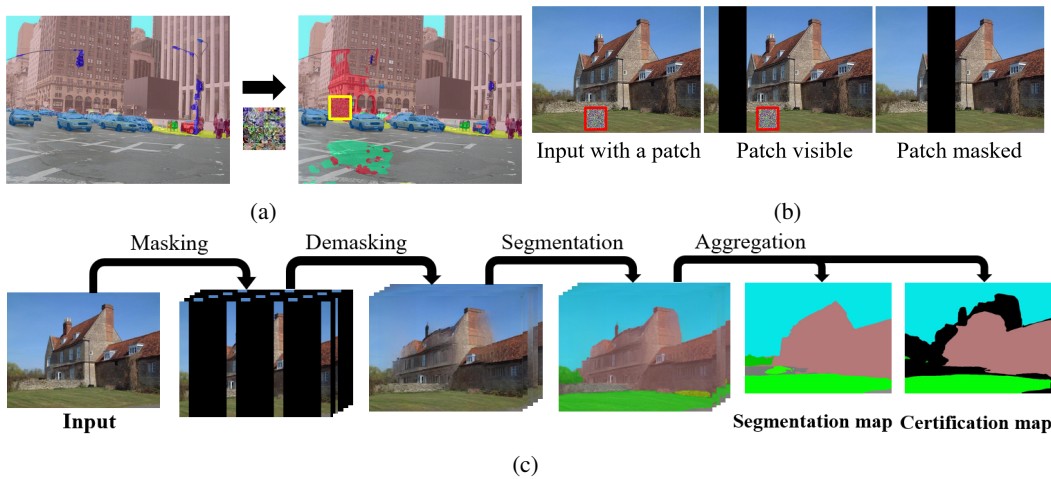

Figure 1: (a) A simple patch attack on the Swin transformer (Liu et al., 2021) manages to switch the prediction for a big part of the image. (b) Masking the patch. (c) A sketch of DEMASKED SMOOTHING for certified image segmentation. First, we generate a set of masked versions of the image such that each possible patch can only affect a certain number of masked images. Then we use image inpainting to partially recover the information lost during masking and then apply an arbitrary segmentation method. The output is obtained by aggregating the segmentations pixelwise. The masking strategy and aggregation method depend on the certification mode (detection or recovery).

Adversarial patch attacks were also proposed for the image segmentation problem (Nesti et al., 2022), mostly for attacking CNN-based models that use a localized receptive field (Zhao et al., 2017). However, recently self-attention based vision transformers (Dosovitskiy et al., 2021) have achieved new state-of-the-art in the image segmentation task (Liu et al., 2021; Bousselham et al., 2021). Their output may become more vulnerable to adversarial patches if they manage to manipulate the global self-attention (Lovisotto et al., 2022). We demonstrate how significant parts of the segmentation output may be affected by a small patch for Swin tranfromer Liu et al. (2021) in Figure 1a. Full details on the attack are available in Appendix D. We point out that preventive certified defences are important because newly developed attacks can immediately be used to compromise safety-critical applications unless they are properly defended.

In this work, we propose the novel framework DEMASKED SMOOTHING (Figure 1c) to obtain the first (to the best of our knowledge) certified defences against patch attacks on semantic segmentation models. Similarly to previous work (Levine & Feizi, 2020), we mask different parts of the input (Figure 1b) and provide guarantees with respect to every possible patch that is not larger than a certain pre-defined size. While prior work required the classification model to deal with such masked inputs, we leverage recent progress in image inpainting (Dong et al., 2022) to reconstruct the input *before* passing it to the downstream model. This decoupling of image demasking from the segmentation task allows us to support arbitrary downstream models. Moreover, we can leverage state of the art methods for image inpainting. We also propose different masking schemes tailored for the segmentation task that provide the dense input allowing the demasking model to understand the scene but still satisfy the guarantees with respect to the adversarial patch.

We summarize our contributions as follows:

- We propose DEMASKED SMOOTHING which is the first (to the best of our knowledge) certified recovery or certified detection based defence against adversarial patch attacks on semantic segmentation models (Section 4).
- DEMASKED SMOOTHING can do certified detection and recovery with any off-the-shelf segmentation model without requiring finetuning or any other adaptation.
- We implement DEMASKED SMOOTHING, evaluate it for different certification objectives and masking schemes (Section 5). We can certify 63% of all pixels in certified detection for a 1% patch and 46% in certified recovery for a 0.5% patch for the BEiT-B (Bao et al., 2022) segmentation model on the ADE20K Zhou et al. (2017) dataset.

## 2 RELATED WORK

**Certified recovery.** The first certified recovery defence for classification models against patches was proposed by Chiang et al. (2020), who adapted interval-bound propagation (Gowal et al., 2019) to the patch threat model. Levine & Feizi (2020) proposed De-Randomized Smoothing (DRS), which provides significant accuracy improvement when compared to Chiang et al. (2020) and scales to the ImageNet dataset. In DRS, a base classifier is trained on images where everything but a small local region is masked (ablated). At inference time, a majority vote of all specified ablations is taken as the final classification. If this vote has a large enough margin to the runner-up class, the prediction cannot be shifted by any patch that does not exceed a pre-defined size. A similar approach was adopted in Randomized Cropping (Lin et al., 2021). A general drawback of these approaches is that the classifier needs to be trained to process masked/cropped inputs, which (in contrast to our work) prohibits the usage of arbitrary pretrained models. A further line of work studies network architectures that are particularly suited for certified recovery. For instance, models with small receptive fields such as *BagNets* (Brendel & Bethge, 2019) have been explored, either by combining them with some fixed postprocessing (Zhang et al., 2020; Xiang et al., 2021) or by training them end-to-end for certified recovery (Metzen & Yatsura, 2021). Salman et al. (2021) propose to apply DRS to *Vision Transfomers (ViTs)*. In contrast to the aforementioned works, our Demasked Smoothing can be applied to models with arbitrary architecture. This is a property shared with PatchCleanser (Xiang et al., 2022a), which however is limited to image classification and it is not clear how it can be extended to semantic segmentation where a class needs to be assigned to every pixel including the masked ones. Certified recovery against patches has also been extended to object detection, specifically to defend against patch hiding attacks. Two notable works in this direction are DetectorGuard (Xiang & Mittal, 2021b), an extension of PatchGuard (Xiang et al., 2021) to object detection, and ObjectSeeker (Xiang et al., 2022b). Randomized smoothing (Cohen et al., 2019) has been applied to certify semantic segmentation models against $\ell_2$-norm bounded adversarial attacks (Fischer et al., 2021). However, to the best of our knowledge, no certified defence against patch attacks for semantic segmentation has been proposed so far.

**Certified detection.** An alternative to certified recovery is *certified detection*. Here, an adversarial patch is allowed to change the model prediction. However, if it succeeds in doing so, there is a mechanism that detects this attack certifiably with zero false negatives. Minority Reports (McCoyd et al., 2020) was the first certified detection method against patches, which is based on sliding a mask over the input in a way that ensures that there will be one mask position that completely hides the patch. PatchGuard++ (Xiang & Mittal, 2021b) is an extension of Minority Reports where the sliding mask is not applied on the input but on the feature maps of a BagNet-type feature extractor. This reduces inference time drastically since the feature extractor needs to be executed only once per input. ScaleCert (Han et al., 2021) tries to identify "superficial important neurons", which allows pruning the network in a way that the prediction needs to be made for fewer masked inputs. Lastly, PatchVeto (Huang & Li, 2021) is a recently proposed method for certified detection that is tailored towards ViT models. It implements masking by removing certain input patches of the ViT. In this work, we propose a novel method for certified detection in the semantic segmentation task that can be used for any pretrained model.

**Image reconstruction.** The problem of learning to reconstruct the full image from inputs where parts have been masked out was pioneered by Vincent et al. (2010). It recently attracted attention as proxy task for self-supervised pre-training, especially for the ViTs (Bao et al., 2022; He et al., 2021). Recent approaches to this problem are using Fourier convolutions (Suvorov et al., 2022) and visual transformers (Dong et al., 2022). SPG-Net (Song et al., 2018) trains a subnetwork to reconstruct the full semantic segmentation directly from the masked input as a part of the image inpainting pipeline. In this work, we use the state-of-the-art ZITS (Dong et al., 2022) inpainting method.

## 3 PROBLEM SETUP

### 3.1 SEMANTIC SEGMENTATION

In this work, we focus on the semantic segmentation task. Let $\mathcal{X}$ be a set of rectangular images. Let $x \in \mathcal{X}$ be an image with height $H$, width $W$ and the number of channels $C$. We denote $\mathcal{Y}$ to be a finite label set. The goal is to find the segmentation map $s \in \mathcal{Y}^{H \times W}$ for $x$. For each pixel $x_{i,j}$, the

corresponding label $s_{i,j}$ denotes the class of the object to which $x_{i,j}$ belongs. We denote $\mathbb{S}$ to be a set of segmentation maps and $f : \mathcal{X} \to \mathbb{S}$ to be a segmentation model.

## 3.2 Threat model

Let us consider an untargeted adversarial patch attack on a segmentation model. Consider an image $x \in [0, 1]^{H \times W \times C}$ and its ground truth segmentation map $s$. Assume that the attacker can modify an arbitrary rectangular region of the image $x$ which has a size of $H' \times W'$. We refer to this modification as a *patch*. Let $l \in \{0, 1\}^{H \times W}$ be a binary mask that defines the patch location in the image in which ones denote the pixels belonging to the patch. Let $\mathcal{L}$ be a set of all possible patch locations for a given image $x$. Let $p \in [0, 1]^{H \times W \times C}$ be the modification itself. Then we define an operator $A$ as $A(x, p, l) = (1 - l) \odot x + l \odot p$, where $\odot$ is element-wise product. The operator $A$ applies the $H' \times W'$ subregion of $p$ defined by a binary mask $l$ to the image $x$ while keeping the rest of the image unchanged. We denote $\mathcal{P} := [0, 1]^{H \times W \times C} \times \mathcal{L}$ to be a set of all possible patch configurations $(p, l)$ that define an $H' \times W'$ patch. Let $s \in \mathbb{S}$ be the ground truth segmentation for $x$. Let $Q(f(x), s)$ be some quality metric such as global pixel accuracy or mean intersection over union (mIoU). The goal of an attacker is to find $(p^\star, l^\star)$ s. t. $(p^\star, l^\star) = \underset{(p, l) \in \mathcal{P}}{\arg \min} \ Q(f(A(x, p, l)), s)$

## 3.3 Defence objective

In this paper, we propose certified defences against patch attacks. It means that we certify against *any possible attack* from $\mathcal{P}$ including $(p^\star, l^\star)$. We consider two robustness objectives.

**Certified recovery** For a pixel $x_{i,j}$ our goal is to verify that the following statement is true

$$\forall (p, l) \in \mathcal{P} : \ f(A(x, p, l))_{i,j} = f(x)_{i,j} \tag{1}$$

**Certified detection** We consider a verification function $v$ defined on $\mathcal{X}$ such that $v(x) \in \{0, 1\}^{H \times W}$. If $v(x)_{i,j} = 1$, then the adversarial patch attack on $x_{i,j}$ can be detected by applying the function $v$ to the attacked image $x' = A(x, p, l)$.

$$v(x)_{i,j} = 1 \Rightarrow \left[ \forall (p, l) \in \mathcal{P} : v(A(x, p, l))_{i,j} = 1 \to f(A(x, p, l))_{i,j} = f(x)_{i,j} \right] \tag{2}$$

$v(x')_{i,j} = 0$ means an alert on pixel $x'_{i,j}$. However, if $x'$ is not an adversarial example, then this is a false alert. In that case the fraction of pixels for which we return false alert is called *false alert ratio* (FAR). The secondary objective is to keep FAR as small as possible.

Depending on the objective our goal is to certify one of the conditions 1, 2 for each pixel $x_{i,j}$. This provides us an upper bound on an attacker's effectiveness under any adversarial patch attack from $\mathcal{P}$.

## 4 Demasked Smoothing

Demasked Smoothing (Figure 1c) consists of several steps. First, we apply a predefined set of masks with specific properties to the input image to obtain a set of masked images. Then we reconstruct the masked regions of each image based on the available information with an inpainting model $g$. After that we apply a segmentation model $f$ to the demasked results. Finally, we aggregate the segmentation outcomes and make a conclusion for the original image with respect to the statements (1) or (2).

### 4.1 Input masking

**Motivation.** Like in previous work (Section 2) we apply masking patterns to the input image and use predictions on masked images to aggregate the robust result. If an adversarial patch is completely masked, it has no effect on further processing. However, in semantic segmentation, we predict not a single whole-image label like in the classification task, but a separate label for each pixel. Thus, making prediction on a masked image must allow us to predict the labels also for the masked pixels.

**Preliminaries.** Consider an image $x \in [0, 1]^{H \times W \times C}$. We define "$*$" to be a special masking symbol that does not correspond to any pixel value and has the property $\forall z \in \mathbb{R} : \ z \times * = *$. Please note that $*$ needs to be different from 0 since 0 is a valid pixel value in unmasked inputs. Let $m \in \{*, 1\}^{H \times W}$ be a *mask*. We call the element-wise product $x \odot m$ a *masking* of $x$. In a masking, a subset of

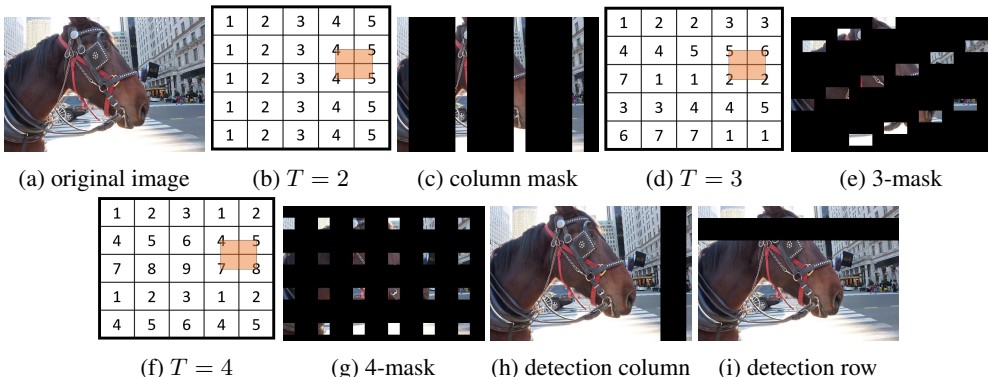

(a) original image    (b) $T = 2$    (c) column mask    (d) $T = 3$    (e) 3-mask

(f) $T = 4$    (g) 4-mask    (h) detection column    (i) detection row

Figure 2: examples of for the column masks: $T = 2$ (b, c), 3-mask: $T = 3$ (d, e), and 4-mask: $T = 4$ (f, g) with the number of masks $K = 5, 7, 9$ respectively. The number on a block denotes in which mask it is visible (there is only one such mask for each block). For each mask set, we show one of the locations $l$ in which an adversarial patch $(p, l)$ affects $T$ different maskings.

pixels becomes $*$ and the rest remains unchanged. We consider the threat model $\mathcal{P}$ with patches of size $H' \times W'$ (Section 3.2). To define the structure of our masks, we break $m$ into an array $B$ of non-intersecting blocks, each having the same size $H' \times W'$ as the adversarial patch. We index the blocks as $B[q, r]$, $1 \leq q \leq \lceil \frac{H}{H'} \rceil$, $1 \leq r \leq \lceil \frac{W}{W'} \rceil$. We say that the block $B[q, r]$ is *visible* in a mask $m$ if $\forall (i, j) \in B[q, r] : m_{i,j} = 1$ Consider an array $M$ of $K$ masks. We define each mask $M[k]$ by a set of blocks that are visible in it. For certified recovery, each block is visible in exactly one mask and masked in the others. We say that a mask $m$ is *affected* by a patch $(p, l)$ if $A(x, p, l) \odot m \neq x \odot m$. We define $T(M) = \max_{(p,l) \in \mathcal{P}} |\{m \in M \,|A(x, p, l) \odot m \neq x \odot m\}|$. That is: $T(M)$ is the largest number of masks affected by some patch. If $M$ is defined, we refer to the value $T(M)$ as $T$ for simplicity.

**Certified recovery.** We define column masking $M$ for which $T = 2$. We assign every $k$-th block column to be visible in the mask $M[k]$ (Figure 2c). Any $(p, l) \in \mathcal{P}$ can intersect at most two adjacent columns since $(p, l)$ has the same width as a column. Thus, it can affect at most two masks (Figure 2b). A similar scheme can be proposed for the rows. Due to the block size in $B$, the patch $(p, l)$ cannot intersect more than four blocks at once. We define a mask set that we call *3-mask* s. t. for any four adjacent blocks two are visible in the same mask (Figures 2d). Hence, a patch for 3-mask can affect no more than 3 masks, $T = 3$. To achieve $T = 4$ any assignment of visible blocks to the masks works. We consider *4-mask* that allows uniform coverage of the visible blocks in the image (Figure 2f). See details on masking schemes in Appendix B.

**Certified detection.** We define $M_d$ to be a set of masks for certified detection (we use subscript $d$ for distinction). $M_d$ should have the property: $\forall (p, l) \in \mathcal{P} \, \exists \, m \in M_d : A(x, p, l) \odot m = x \odot m$ i. e. for every patch exists at least one mask not affected by this patch. For a patch of size $H' \times W'$ we consider $K = W - W' + 1$ masks such that the mask $M_d[k]$ masks a column of width $W'$ starting at the horizontal position $k$ in the image (Figure 2h). To obtain the guarantee for the same $\mathcal{P}$ with a smaller $K$, we consider a set of strided columns of width $W'' \geq W'$ and stride $W'' - W' + 1$ that also satisfy the condition (see the proof adapted from Xiang et al. (2022a) in Appendix A). A similar scheme can be proposed for the rows (Figure 2i). Alternatively, we could use a set of block masks of size $H' \times W'$. Then the number of masks grows quadratically with the image resolution. Hence, in the experiments we focus on the column and the row masking schemes.

Let $g$ be a demasking model, $g(x \odot m) \in [0, 1]^{H \times W \times C}$. The goal of $g$ is to make the reconstruction $g(x \odot m)$ as close as possible (in some metric) to the original image $x$. For a segmentation model $f$ we define a *segmentation array* $S(M, x, g, f)$, $S[k] := f(g(x \odot M[k]))$, $1 \leq k \leq K$.

## 4.2 CERTIFICATION

**Certified recovery.** For the threat model $\mathcal{P}$ consider a set $M$ of $K$ masks. We define a function $h : \mathcal{X} \to \mathbb{S}$ that assigns a class to the pixel $x_{i,j}$ via majority voting over class predictions of each

---

**Algorithm 1** Demasked Smoothing

---

**Input:** image $x \in [0, 1]^{H \times W \times C}$, patch size $(H', W')$, certification type CT (recovery or detection), mask type MT (column, row, 3-mask, 4-mask), inpainting model $g$, segmentation model $f$
**Output:** segmentation map $h \in \mathcal{Y}^{H \times W}$, certification (or verification) map $v \in \{0, 1\}^{H \times W}$

  1:  $M \leftarrow \text{CreateMaskArray}(H, W, H', W', \text{CT}, \text{MT})$       ▷ according to section 4.1
  2: **for** $k \leftarrow 1, \ldots, |M|$ **do**                 ▷ this loop can be paralellized
  3:     $S[k] \leftarrow f(g(x \odot M[k]))$ ▷ mask input, inpaint the masked regions, and apply segmentation
  4: **end for**
  5: **if** CT = 'recovery' **then** $h \leftarrow \text{MajorityVote}(S)$  ▷ vote over the classes predicted for each pixel
  6: **else**    $h \leftarrow f(x)$                ▷ in detection case, output clean segmentation
  7: **end if**
  8: $v \leftarrow \text{AllEqual}(S, h)$      ▷ assign 1 for the pixels where all $S[k]$ agree with $h$ and 0 otherwise
  9: **Return** $h, v$

---

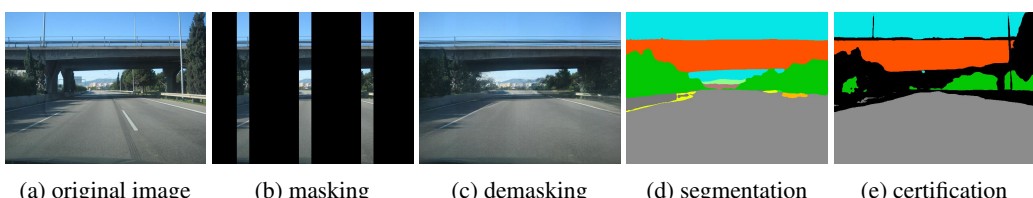

(a) original image      (b) masking      (c) demasking      (d) segmentation      (e) certification

Figure 3: Reconstructing the masked images with ZITS Dong et al. (2022)

reconstructed segmentation in $S$. A class for the pixel that is predicted by the largest number of segmentations is assigned. We break the ties by assigning a class with a smaller index.

**Theorem 1.** *If the number of masks $K$ satisfies $K \geq 2T(M) + 1$ and for a pixel $x_{i,j}$ we have*

$$\forall \, S[k] \in S : \, S[k]_{i,j} = h(x)_{i,j}$$

*(i.e. all the votes agree), then $\forall \, (p, \, l) \in \mathcal{P} : \, h(A(x, p, l))_{i,j} = h(x)_{i,j}$.*

See the proof in Appendix A.

**Certified detection.** Consider $M_d = \{M_d[k]\}_{k=1}^{K}$. For a set of demasked segmentations S we define the verification map $v(x)_{i,j} := [f(x)_{i,j} = S[1]_{i,j} = \ldots = S[K]_{i,j}]$ i.e. the original segmentation is equal to all the other segmentations on masked-demasked inputs, including the one in which the potential patch was completely masked.

**Theorem 2.** *Assume that $v(x)_{i,j} = 1$. Then*

$$\forall \, (p, \, l) \in \mathcal{P} : v(A(x, p, l))_{i,j} = 1 \Rightarrow f(A(x, \, p, \, l))_{i,j} = f(x)_{i,j}$$

See the proof in Appendix A. For a given image $x$ the verification map $v(x)$ is complementary to the model segmentation output $f(x)$ that stays unchanged. Thus, there is no drop in clean performance however we may have some false positive alerts in the verification map $v$ in the clean setting. We present the Demasked Smoothing procedure in Algorithm 1.

## 5   EXPERIMENTS

In this section, we evaluate DEMASKED SMOOTHING with the masking schemes proposed in Section 4, compare our approach with the direct application of Derandomized Smoothing Levine & Feizi (2020) to the segmentation task and evaluate the performance on different datasets and models. Certified recovery and certified detection provide certificates of different strength (Section 4) which are not comparable. We evaluate them separately for different patch sizes.

### 5.1   EXPERIMENTAL SETUP

We evaluate DEMASKED SMOOTHING on two challenging semantic segmentation datasets: ADE20K (Zhou et al., 2017) (150 classes, 2000 validation images) and COCO-Stuff-10K (Caesar et al., 2018)

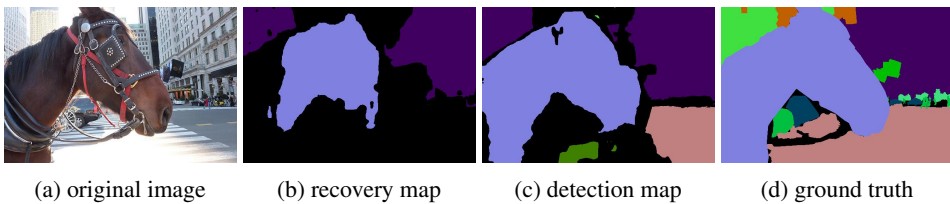

| (a) original image | (b) recovery map | (c) detection map | (d) ground truth |

Figure 4: Certification maps for recovery and detection

(171 classes, 1000 validation images). For demasking we use the ZITS Dong et al. (2022) inpainting model with the checkpoint provided in the official paper repository [1]. The model was trained on Places2 (Zhou et al., 2016) dataset with images resized to 256×256. As a segmentation model $f$ we use BEiT-B Bao et al. (2022), Swin Liu et al. (2021), PSPNet Zhao et al. (2017) and DeepLab v3 (Chen et al., 2018). We note that the first two models are based on transformers and obtain near state-of-the-art results. PSPNet and DeepLab v3 are CNN-based segmentation methods that we consider to demonstrate that DEMASKED SMOOTHING is not specific to transformer-based architectures. We use the model implementations provided in the *mmsegmentation* framework Contributors (2020). An illustration of the image reconstruction and respective segmentation can be found in Figure 3. We run the evaluation in parallel on 5 Nvidia Tesla V100-32GB GPUs. The certification for the whole ADE20K validation set with ZITS and BEiT-B takes around 1.2 hours for certified recovery and 2 hours for certified detection (due to a larger number of masks).

## 5.2 EVALUATION METRICS

For both certified recovery and certified detection, we generate a standard segmentation output (without any abstention) and a corresponding certification map (Figure 4). In case of certified detection, the segmentation output remains the same as for the original segmentation model, however, there may be false alerts in the certificaton map. For the certified recovery, the output is obtained by a majority vote over the segmentations of demasked images (Section 4.2). We evaluate the mean intersection over union (mIoU) for these outputs. The certification map is obtained by assigning to each certified pixel the corresponding class from the segmentation output and assigning a special *uncertified* label to all non-certified pixels. For each image we evaluate the fraction of pixels which are certified and correct (coincide with the ground truth). %C is a mean of these fractions over all the images in the dataset. In semantic segmentation task, the class frequencies are usually skewed, therefore global pixel-wise accuracy alone is an insufficient metric.

Matching the certification map separately for each class $y \in \mathcal{Y}$ with the ground truth segmentation for $y$ in the image $x$ allows us to compute the guaranteed lower bound ($cTP_y(x)$) on the number of true positive pixel predictions ($TP_y(x)$) i.e. those that were correctly classified into $y$. If a pixel was certified with a correct class, then this prediction cannot be changed by a patch (or, alternatively, the change will be detected by the verification function $v$ in certified detection). We consider *recall* $R_y(x) = \frac{TP_y(x)}{TP_y(x)+FN_y(x)}$ where $FN_y(x)$ is the number of false negative predictions for $y$ in $x$. $P_y(x) = TP_y(x) + FN_y(x)$ is the total area of $y$ in the ground truth and does not depend on our prediction. We can evaluate certified recall $cR_y(x) = \frac{cTP_y(x)}{P_y(x)}$, a lower bound on the recall $R_y(x)$. Total recall and certified total recall of class $y$ in a dataset $D$ are $TR_y(D) = \frac{\sum_{x \in D} TP_y(x)}{\sum_{x \in D} P_y(x)}$ and $cTR_y(D) = \frac{\sum_{x \in D} cTP_y(x)}{\sum_{x \in D} P_y(x)}$ respectively. Then, we obtain mean recall $mR(D) = \frac{1}{|\mathcal{Y}|} \sum_{y \in \mathcal{Y}} TR_y(D)$ and certified mean recall $cmR(D) = \frac{1}{|\mathcal{Y}|} \sum_{y \in \mathcal{Y}} cTR_y(D)$. Evaluating lower bounds on other popular metrics such as mean precision or mIoU this way results in vacuous upper bound since they depend on the upper bound on false positive ($FP$) predictions. For the pixels that are not certified we cannot guarantee that they will not be assigned to a certain class, therefore, a non-trivial upper bound on $FP$ is not straightforward. We leave this direction for future work. In certified detection, we additionally consider false alert ratio (FAR) which is the fraction of correctly classified pixels for which we return an alert on a clean image. Smaller FAR is preferable.

---

[1] https://github.com/DQiaole/ZITS_inpainting

Table 1: Comparison of different masking schemes proposed in Section 4.1. mIoU - mean intersection over union, mR - mean recall, cmR - certified mean recall. %C - mean percentage of certified and correct pixels in the image. For detection, we provide clean mIoU since the output is unaffected and mean false alert rate (FAR) (lower is better). See additional results in Appendix E.

| mode | dataset | segm | mask | mIoU | big mR | big cmR | all mR | all cmR | %C | FAR $\downarrow$ |
|---|---|---|---|---|---|---|---|---|---|---|
| detection 1% patch | ADE20K | BEiT-B | column | 53.08 | 70.92 | **57.33** | 64.45 | **32.55** | **63.55** | **20.04** |
| | | | row | | | 50.05 | | 26.65 | 58.34 | 25.24 |
| | COCO10K | PSPNet | column | 37.76 | 71.71 | **56.86** | 49.65 | **26.80** | **47.09** | **21.43** |
| | | | row | | | 51.05 | | 23.51 | 42.78 | 25.74 |
| recovery 0.5% patch | ADE20K | BEiT-B | column | **24.92** | **60.77** | **41.26** | **29.84** | **12.98** | **46.22** | |
| | | | row | 16.33 | 46.91 | 16.72 | 19.51 | 4.83 | 31.71 | |
| | | | 3-mask | 19.90 | 56.90 | 26.51 | 23.86 | 7.54 | 38.64 | |
| | | | 4-mask | 18.82 | 52.96 | 23.75 | 22.56 | 5.87 | 34.36 | N/A |
| | COCO10K | PSPNet | column | **21.94** | **61.56** | **36.67** | **29.94** | **11.13** | **29.51** | |
| | | | row | 18.87 | 58.04 | 20.90 | 26.16 | 6.14 | 19.31 | |
| | | | 3-mask | 18.82 | 59.26 | 29.00 | 25.85 | 7.56 | 25.21 | |
| | | | 4-mask | 17.46 | 58.47 | 23.63 | 24.35 | 5.51 | 20.36 | |

Table 2: Comparison of our method with an adaptation of Levine & Feizi (2020) to segmentation. We use Swin Liu et al. (2021) on 200 ADE20K images. See details and illustrations in Appendix G.

| method | mIoU | big mR | big cmR | all mR | all cmR | %C |
|---|---|---|---|---|---|---|
| Demasked Smoothing (our) | **19.09** | **66.03** | **52.71** | **23.02** | **12.66** | **47.05** |
| DRS-S | 0.42 | 11.35 | 9.08 | 1.04 | 0.83 | 28.01 |
| DRS-E | 9.12 | 54.67 | 41.78 | 11.04 | 7.86 | 45.03 |

Due to our threat model, certifying small objects in the scene can be difficult because they can be partially or completely covered by an adversarsial patch in a way that there is not chance to recover the prediction. To provide an additional perspective on our methods, we also evaluate mR and cmR specifically for the "big" classes, which occupy on average more than 20% of the images in which they appear. These are, for example, road, building, train, and sky, which are important for understanding the scene. The full list of such classes for each dataset is provided in the Appendix H.

## 5.3 DISCUSSION

In Table 1, we compare different masking schemes proposed in Section 4.1. Evaluation of all the models with all the masking schemes is consistent with these results and can be found in Appendix E. We see that column masking achieves better results in both certification modes. Effectiveness of column masking for classification task was also empirically observed by Levine & Feizi (2020). We attribute the effectiveness of column masking to the fact most of the images in the datasets have a clear horizont line, therefore having a visible column provides a slice of the image that intersects most of the scene background objects.

In Table 2, we extend Derandomized Smoothing (DRS) proposed by Levine & Feizi (2020) for certified recovery in classification to the segmentation task and compare it to our method Direct adaptation requires training a model that is able to predict the full image segmentation from a small visible region. Since it is not clear what architectural design and training procedure would be needed for that, we consider two alternative baselines. DRS-S predicts the segmentation directly from the masked image and DRS-E uses our inpainting method to first reconstruct the image and then obtain the segmentation. See the implementation details in Appendix G. DRS with column smoothing performs poorly on the segmentation task, which emphasizes the need for specific masking schemes.

In Table 3, we evaluate our method with column masking on different models. For certified detection we can certify more than 60% of the pixels with all models on ADE20K and more than 46 % on

Table 3: Demasked Smoothing results with column masking for different models

| mode | dataset | segm | mIoU | big mR | big cmR | all mR | all cmR | %C | FAR ↓ |
|------|---------|------|------|-----|------|-----|------|-----|-------|
| detection 1 % patch | ADE20K | BEiT-B | **53.08** | **70.92** | **57.33** | **64.45** | **32.55** | **63.55** | 20.04 |
| | | PSPNet | 44.39 | 61.83 | 50.02 | 54.74 | 26.37 | 60.57 | 20.08 |
| | | Swin-B | 48.13 | 68.51 | 55.45 | 59.13 | 29.06 | 61.44 | 20.31 |
| | COCO10K | PSPNet | 37.76 | 71.71 | 56.86 | 49.65 | 26.80 | 47.09 | 21.43 |
| | | DeepLab v3 | 37.81 | 72.52 | 56.54 | 49.98 | 26.86 | 46.55 | 21.89 |
| recovery 0.5 % patch | ADE20K | BEiT-B | **24.92** | **60.77** | **41.26** | **29.84** | **12.98** | **46.22** | N/A |
| | | PSPNet | 19.17 | 51.90 | 34.11 | 23.66 | 10.76 | 44.90 | |
| | | Swin-B | 22.43 | 59.75 | 34.88 | 27.09 | 11.70 | 46.14 | |
| | COCO10K | PSPNet | 21.94 | 61.56 | 36.67 | 29.94 | 11.13 | 29.51 | |
| | | DeepLab v3 | 23.12 | 62.60 | 33.84 | 31.59 | 11.55 | 28.71 | |

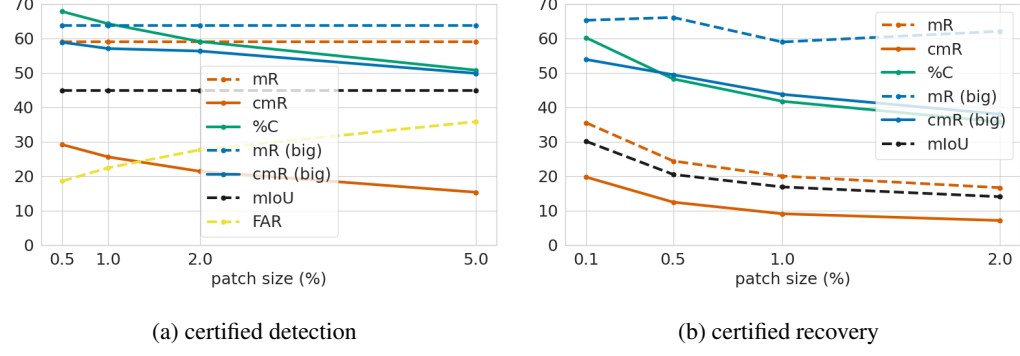

(a) certified detection          (b) certified recovery

Figure 5: Performance for different adversarial patch sizes evaluated on 200 ADE20K images.

COCO10K. False alert ratio on correctly classified pixels is around 20%. In certified recovery, we certify more than 44% pixels on ADE20K and more than 28% pixels on COCO10K. Figure 5 shows how the performance of DEMASKED SMOOTHING depends on the patch size for the BEiT-B model. We see that certified detection metrics remain high even for a patch as big as 5% of the image surface and for the recovery they slowly deteriorate as we increase the patch size to 2%. Ablations with respect to inpainting can be found in Appendix F. DEMASKEDSMOOTHING illustrations procedure are provided in Appendix K.

### 5.4 LIMITATIONS

The performance of Demasked Smoothing certified recovery may be insufficient for the downstream task if we certify against big patches (Figure 5) unless robustness is prioritized over clean performance. We point out that robustly segmenting small objects is fundamentally difficult under the adversarial patch threat model since the objects themselves can be completely or partially covered by an adversarial patch which makes it impossible to properly segment them even for a human being. Demasked Smoothing certification requires an upper bound on the expected patch size (Section 4).

### 6 CONCLUSION

In this work, we propose DEMASKED SMOOTHING, the first (up to our knowledge) certified defence framework against patch attacks on segmentation models. Due to its novel design based on masking schemes and image demasking, DEMASKED SMOOTHING is compatible with any segmentation model and can on average certify 63% of the pixel predictions for a 1% patch in the detection task and 46% against a 0.5% patch for the recovery task on the ADE20K dataset.

## 7 ETHICS STATEMENT

This work contributes to the field of certified defences against physically-realizable adversarial attacks. The proposed approach allows to certify robustness of safety-critical applications such as medical imaging or autonomous driving. The defence might be used to improve robustness of systems used for malicious purposes such as (semi-)autonomous weaponry or unauthorized surveillance. This danger may be mitigated e.g. by using a system of sparsely distributed patches which makes certifying the image more challenging. All activities in our organization are carbon neutral, so the experiments performed on our GPUs do not leave any carbon dioxide footprint.

## 8 REPRODUCIBILITY STATEMENT

We provide the details of our experimental setup in Section 5.1. We discuss the evaluation metrics and their computation in Section 5.2.

### ACKNOWLEDGEMENTS

We thank Chong Xiang for the suggestions on extending our evaluation protocol. Matthias Hein is a member of the Machine Learning Cluster of Excellence, EXC number 2064/1 – Project number 390727645 and acknowledges support of the Carl Zeiss Foundation in the project "Certification and Foundations of Safe Machine Learning Systems in Healthcare"

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

# A    PROOFS (SECTION 4)

In this section, we provide the proofs for the theorems stated in Section 4.

**Lemma 1.** *(Section 4.1) Consider an image of the size $H \times W$. Let $H' \times W'$ be a fixed adversarial patch size. Let $M^d$ be a set of $K$ masks where each mask is masking an $H \times W''$ vertical column, $W'' \geq W'$. Let the stride between the columns in two adjacent masks be $W'' - W' + 1$. Then for any location $l \in \mathcal{L}$ of the patch, there exists a mask that covers it completely.*

*Proof.* (Adapted from the proof of Lemma 4 in PatchCleanser Xiang et al. (2022a)). Without loss of generality, we consider the first two adjacent column masks. The first one covers the columns from 1 to $W''$. The second mask covers the columns from $1 + (W'' - W' + 1) = W'' - W' + 2$ to $(W'' - W' + 2) + (W'' - 1) = 2W'' - W' + 1$ (See Figure 6). Now consider an adversarial patch of size $H' \times W'$. Let us find the smallest possible start index of this patch so that it does not get covered by the first mask. For that it should be visible at the column $W'' + 1$ and, therefore, start at the column with index not smaller than $(W'' + 1) - W' + 1 = W'' - W' + 2$. However, it is the same column in which second mask starts. Therefore, given that $W'' \geq W'$ we have that the patch is completely masked by the second mask. Then for a patch which is only partially masked by the second mask from the left we use an analogous argument to show that it is completely masked by the third mask and so on. $\square$

**Certified recovery.** For the threat model $\mathcal{P}$ (Section 3) consider a set $M$ of $K$ masks. We define a function $h : \mathcal{X} \to \mathbb{S}$ that assigns a class to the pixel $x_{i,j}$ via majority voting over class predictions of each reconstructed segmentation in $S$. A class for the pixel that is predicted by the largest number of segmentations is assigned. We break the ties by assigning a class with a smaller index.

**Theorem 1**. (Section 4.2) If the number of masks $K$ satisfies $K \geq 2T(M) + 1$ and for a pixel $x_{i,j}$ we have

$$\forall\, S[k] \in S : \; S[k]_{i,j} = h(x)_{i,j}$$

i.e. all the votes agree, then $\forall\, (p,\, l) \in \mathcal{P} : \; h(A(x, p, l))_{i,j} = h(x)_{i,j}$.

*Proof.* We prove the statement by contradiction. Assume that

$$\exists\, (p,\, l) \in \mathcal{P} : \; h(A(x, p, l))_{i,j} \neq h(x)_{i,j}$$

Let us denote $x' := A(x, p, l)$ and $S'$ to be the segmentation array for $x'$. We denote the class $h(x)_{i,j}$ predicted for the pixel $x_{i,j}$ as $C$. $h(A(x, p, l))_{i,j} \neq C$ means that the class $C$ did not get the majority in the votes over the segmentation array $S'$. However, by definition of $T(M)$ we know that $(p,\, l)$ could affect at most $T(M)$ segmentations out of $K$ and change their vote. Since all $K$ segmentations of $S$ have voted for $C$, then at least $K - T(M)$ of them are still voting for $C$ in $S'$. And by our assumption $K \geq 2T(M) + 1$, we have that $K - T(M) \geq T(M) + 1$. Thus, the class $C$ for $x_{i,j}$ still gets the majority vote in $S'$. Therefore $h(x')_{i,j} = C = h(x)_{i,j}$. We have arrived to a contradiction. $\square$

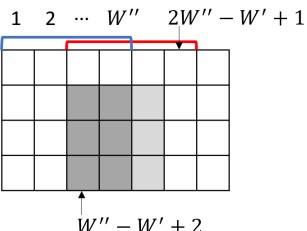

Figure 6: The masked columns of the first two adjacent masks (blue for the first one and red for the second one). If the patch is not completely masked by the first mask, it should be visible at the column $W'' + 1$ (the masked part of the patch is dark-grey and the visible part is in light-grey). However then the patch will be completely masked by the second mask.

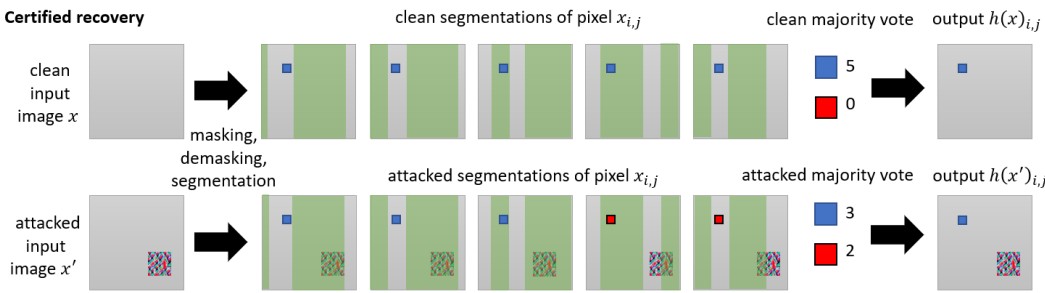

Figure 7: Illustration for Theorem 1 with $K = 5$ masks and $T = 2$. If the majority voting across the five masks is univocal for the clean image, the patch can only affect two out of five segmentations and, thus, cannot shift the majority. The green shade schematically shows which parts were masked in the masking step.

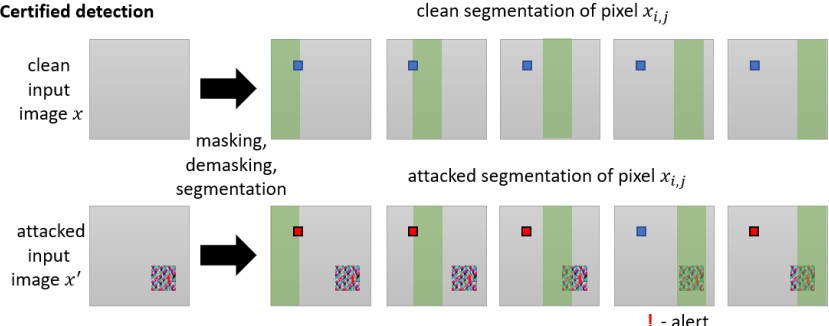

Figure 8: Illustration for Theorem 2. There exists at least one segmentation for which the patch was masked. If the patch has managed to affect the segmentations in which it is not masked, there will be an inconsistency. The green shade schematically shows which parts were masked in the masking step.

A schematic illustration for the certified detection mechanism is provided in Figure 7.

**Certified detection.** Consider $M_d = \{M_d[k]\}_{k=1}^{K}$. For a set of demasked segmentations S we define the verification map $v(x)_{i,j} := [f(x)_{i,j} = S[1]_{i,j} = \ldots = S[K]_{i,j}]$ i.e. the original segmentation coincides with all the other segmentations including the one in which the potential patch was completely masked.

**Theorem 2**. (Section 4.2) Assume that $v(x)_{i,j} = 1$. Then

$$\forall\, (p,\, l) \in \mathcal{P} : v(A(x, p, l))_{i,j} = 1 \Rightarrow f(A(x,\, p,\, l))_{i,j} = f(x)_{i,j}$$

*Proof.* We prove the statement by contradiction. Assume that

$$\exists\, (p,\, l) \in \mathcal{P} :\ v(A(x, p, l))_{i,j} = 1 \wedge f(A(x,\, p,\, l))_{i,j} \neq f(x)_{i,j}$$

Let us denote $x' := A(x, p, l)$ and $S'$ to be the segmentation set for $x'$. By definition of $M_d$, $\exists\, M_d[k] \in M_d$ s. t. $M_d[k]$ masks the patch $(p,\, l)$ Hence,

$$g(x \odot M_d[k]) = g(x' \odot M_d[k]),$$

$$S[k] = f(g(x \odot M_d[k])) = f(g(x' \odot M_d[k])) = S'[k],$$

Since $v(x)_{i,j} = 1$, we have $f(x)_{i,j} = S[k]_{i,j}$. Since $v(x')_{i,j} = 1$, we have $f(x')_{i,j} = S'[k]_{i,j}$. Thus, $f(x')_{i,j} = f(x)_{i,j}$. We have arrived to a contradiction. $\square$

A schematic illustration for the certified recovery mechanism is provided in Figure 8.

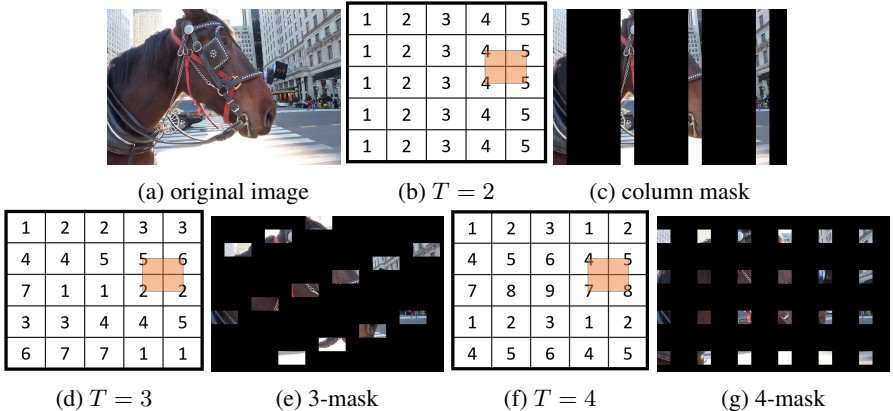

Figure 9: (a) examples of a mask for the column masks with $T = 2$ (b, c), 3-mask with $T = 3$ (d, e), and 4-mask with $T = 4$ (f, g) with the number of masks $K = 5, 7, 9$ respectively. The number of a block denotes in which mask it is not masked (there is only one such mask for each block). For each mask set, we show one of the locations $l$ in which an adversarial patch affects $T$ different maskings.

## B  DETAILED DESCRIPTION OF MASKING STRATEGIES

In this section, we provide additional details for constructing certified recovery masks proposed in Section 4.1. We define mask sets $M$ that satisfy different values of $T$. We divide the image $x$ into a set of non-intersecting blocks $B$ of the same size as an adversarial patch, $H' \times W'$ (see Figure 9), $1 \leq q \leq \lceil H/H' \rceil, 1 \leq r \leq \lceil W/W' \rceil$. In each mask, each of these blocks will be either masked or not masked (i. e. *visible*). Moreover, for each block there exists only one mask in which it is visible. For a set $M$ of $K$ masks we define the mapping $\mu_M : B \to \{1, \dots, K\}$. If $\mu(B[q, r]) = k$, then $B[q, r]$ is not masked in $M[k]$. Therefore, each mask $M[k]$ is defined by a $B_k \subset B$ s. t. for $b \in B_k$ $\mu(b) = k$.

We define a set $M$ that we call 3-mask for which $T(M) = 3$. We assign the blocks in each row to the masks as follows: $\mu(B[1, 1]) = 1; \mu(B[1, 2]) = \mu(B[1, 3]) = 2; \mu(B[1, 4]) = \mu(B[1, 5]) = 3$ and so on until we reach the end of the row. If we finish the first row with the value $k$, then we start the second row as follows $\mu(B[2, 1]) = \mu(B[2, 2]) = k + 1; \mu(B[2, 3]) = \mu(B[2, 4]) = k + 2$: . . .. If we finish the second row on $n$, we start the third row similarly to the first: $\mu(B[3, 1]) = n + 1$; $\mu(B[3, 2]) = \mu(B[3, 3]) = n + 2; \dots$ When we reach the number $K$, we start from 1 again (Figure 9d). Due to the block size, the patch cannot intersect more than four blocks at once. Our parity-alternating block sequence ensures that in any such intersection of four blocks either the top ones or the bottom ones will belong to the same masking, so at most three different maskings can be affected.

We define a set $M$ that we call 4-mask for which $T(M) = 4$. Due to our block size any assignment of masks will work because the patch cannot intersect more than four blocks. We consider the one that allows uniform distribution of the unmasked blocks (Figure 9g). We point out that for the described methods each masking keeps approximately $1/K$ of the pixels visible and the unmasked regions are uniformly distributed in the image. This means that for any masked pixel there exists an unmasked region located close enough to this pixel. It is the core difference between our masks and the ones proposed for certified classification such as block or column smoothing Levine & Feizi (2020). It was observed that the image demasking is facilitated when the visible regions are uniformly spread in the masked image He et al. (2021).

## C  TEST-TIME INPUT CERTIFICATION

In this section, we discuss how certified recovery (Theorem 1) can be applied to guaranteed verification of the robustness on a test image. We also discuss how robustness guarantees for the test-time images can be evaluated by using a dataset of clean images such as ADE20K (Zhou et al., 2017) or COCO-Stuff-10K (Caesar et al., 2018).

## C.1 Test-time certified recovery

Let $x'$ be a test-time input which can be either a clean image or an image attacked with an adversarial patch. We know that there exists a clean image $x$ corresponding to $x'$ which removes the patch if it is present. We have either $x' = x$ or $x' \in A(x)$, where $A(x) := \{A(x, p, l) \mid (p, l) \in \mathcal{P}\}$. However, at test time we do not have access to the clean image $x$.

Our goal is to certify that for our segmentation model $h$ and a pixel $x_{i,j}$ we have $h(x')_{i,j} = h(x)_{i,j}$. We can achieve this result by applying the recovery certification (Theorem 1) to the test-time image. It allows us to verify whether $\forall\, (p,\ l) \in \mathcal{P}\ :\ h(A(x', p, l))_{i,j} = h(x')_{i,j}$. We also know that if $x' \in A(x)$, then $x \in A(x')$ (Figure 10a). Indeed, if $x'$ is only different from $x$ by one patch, then $x$ can be be obtained from $x'$ by removing this patch. Therefore, by obtaining the guarantee for $A(x')$, we implicitly obtain the guarantee also for the image $x$ even though we do not have direct access to it.

We note that this test-time guarantee is only possible for certified recovery. In certified detection, we would need to evaluate the verification function $v$ (Theorem 2) for both the clean image $x$ and the attacked image $x'$ to obtain the result. This cannot be done if $x$ is implicit.

## C.2 Robustness guarantees evaluation

The typical certified robust error for a given test data set (and pixel $(i, j)$ in the segmentation case) is an estimate for

$$\mathbb{E}_{X \sim D}\big[\max_{(p,l) \in P} \mathbb{1}_{h(A(X,p,l))_{i,j} \neq h(X)_{i,j}}\big],$$

where $D$ is the data generating probability measure and we assume that our test set to be an i.i.d. sample of it. This is the expected robust error (worst case over our threat model $P$ for clean inputs) for a given pixel $(i, j)$. Using the test sample to get an estimate of this quantity, we get a probabilistic guarantee that the corresponding pixel $(i, j)$ of a new *clean* test sample $x'$ drawn i.i.d. from $P$ will have its whole "patch"-neighborhood certified.

However, more important for a practical security analysis is that we can certify a given instance, which can be even potentially adversarially perturbed. Formally, this means that for an input $z \in A(x)$, where $x \sim P$ is an unknown sample from $P$, that we guarantee

$$\forall (p, l) \in P : h(A(z, p, l))_{i,j} = h(z)_{i,j},$$

and as $x \in A(z, p, l)$ this implies that we certify that the pixel $(i, j)$ of the potentially manipulated image is classified the same as pixel $(i, j)$ of the unperturbed image $x$.

However, it is now tricky to get even a probabilistic estimate of the quantity

$$\mathbb{E}_{x \sim D} \max_{(p,l) \in P}\big[\max_{(q,m) \in P} \mathbb{1}_{h(A(A(x,p,l),q,m))_{i,j} = h(A(x,p,l))_{i,j}}\big],$$

as the outer maximization process cannot be simply simulated by doing adversarial patch attacks on a clean test dataset.

We propose a way to evaluate a guaranteed lower bound on the fraction of certified test-time inputs by using a dataset of clean images. Instead of considering a standard one-patch neighbourhood $A(x)$ defined by our threat model (Section 3.2), we propose to consider a neighbourhood $A^2(x)$ of two independent patches (Figure 10b). $A^2(x)$ contains all the images $x' \in A(x)$ as well as their respective patch neighbourhoods $A(x')$. Therefore, by verifying that $\forall\, (p_1,\ l_1), (p_2,\ l_2) \in \mathcal{P}\ :\ h(A(A(x,\ p_1,\ l_1),\ p_2,\ l_2))_{i,j} = h(x)_{i,j}$, we guarantee that $\forall\, x' \in A(x)\ \forall\, (p,\ l) \in \mathcal{P}\ :\ h(A(x',\ p,\ l))_{i,j} = h(x')_{i,j}$.

We note that corresponding reasoning could be applied to certification in $\ell_p$ models. Then $A^2(x)$ would correspond to doubling the radius of the $\epsilon$-ball instead of adding a second patch.

Note that Theorem 1 can be directly extended to a threat model of $N$ patches. In the worst case each of the $N$ patches can affect $T$ different maskings. Therefore, we need to change the condition of Theorem 1 to $K \geq 2NT + 1$. We apply the described method to evaluating the test-time certification guarantees for a toy example of a $0.1\%$ patch in Table 4. We also illustrate how a column mask looks in this case in Figure 10.

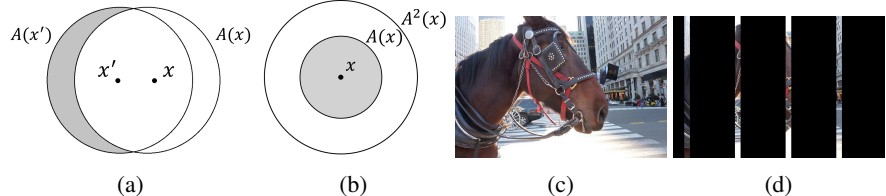

Figure 10: (a) certified inference; (b) double adversarial neighbourhood; (c) original image (d) certification against two patches

Table 4: Inference recovery robustness estimate. To illustrate our point we certify an example for a 0.1% patch.

| dataset | segm | mask | mIoU | big | | all | | %C |
|---|---|---|---|---|---|---|---|---|
| | | | | mR | cmR | mR | cmR | |
| ADE20K | BEiT-B | col | 19.73 | 36.95 | 16.64 | 24.23 | 9.24 | 41.96 |
| COCO10K | | | 26.36 | 69.63 | 35.34 | 34.92 | 11.13 | 28.17 |

# D   ADVERSARIAL PATCH EXAMPLE

In this section, we demonstrate an example of a real adversarial patch for a semantic segmentation model similar to the one illustrated in the Figure 1a and show how it is handled by our certified defences. We illustrate it for the Swin (Liu et al., 2021) model on one of the images from the ADE20K (Zhou et al., 2017) dataset.

## D.1   PATCH OPTIMIZATION

We set the patch size to 1% of the image surface. We select a fixed position for a patch on the rear window of a car (Figure 11a). For each pixel we extract a list of predicted logits corresponding to each class and apply multi-margin loss with respect to the ground truth label of the respective pixel. We use random patch initialization without restarts. As an optimizer we use projected gradient descent (PGD) with 1000 steps and initial step size of 0.01. We use cosine step size schedule and momentum for the gradient with the rate of 0.9. The optimization plot and the patch efficiency at different iterations of the PGD are illustrated in the Figure 11.

## D.2   CERTIFIED RECOVERY

We denote the original image as $x$ and the patched image as $x'$. The voting-based segmentation function $h$ (Section 4.2) provides the majority-vote prediction $h(x)$ and the corresponding certification map which shows the pixels where all the votes agree. In Figure we see that a part of the building and the road is certified which means that this prediction cannot be affected by an adversarial patch. Figure demonstrates $h(x')$ which correctly segments those regions in presence of an adversarial patch that fools the original model.

## D.3   CERTIFIED DETECTION

We perform our analysis by evaluating the verification map $v$ (Section 4.2) for the original image $x$ and for the patched image $x'$. We see that in $v(x)$ a major part of the building is certified i. e. for a part of pixels $x_{i,j}$ that belong to the building and the road we have $v(x)_{i,j} = 1$. However, $v(x')_{i,j} = 0$ for those pixels. It means that we have detected that the prediction on this input is potentially affected by an adversarial patch.

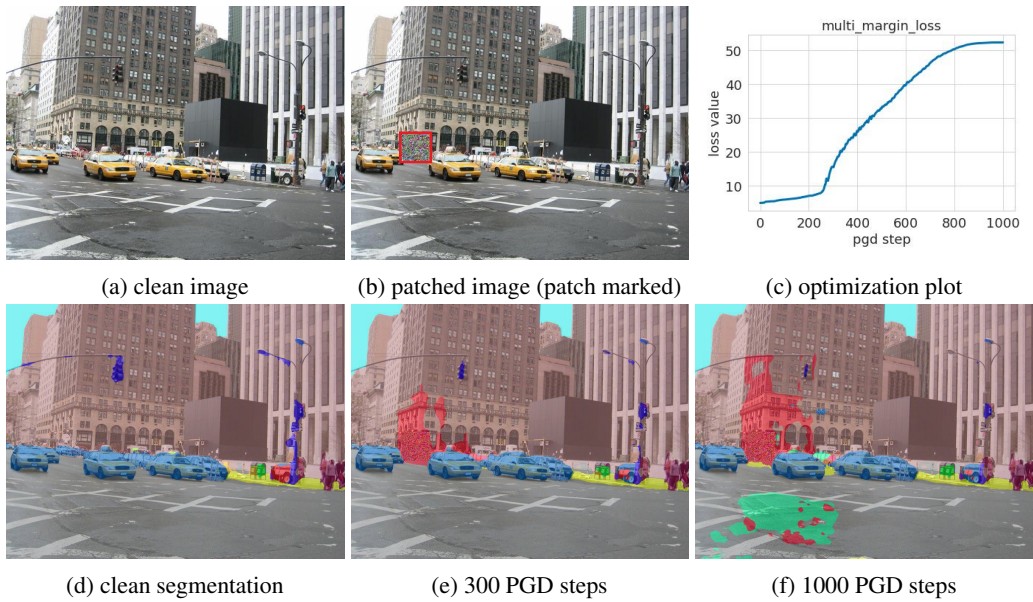

| (a) clean image | (b) patched image (patch marked) | (c) optimization plot |
|---|---|---|
| (d) clean segmentation | (e) 300 PGD steps | (f) 1000 PGD steps |

Figure 11: Patch attack illustration with Swin (Liu et al., 2021) and an ADE20K image. A patch occupying 1% of the image surface changes the segmentation.

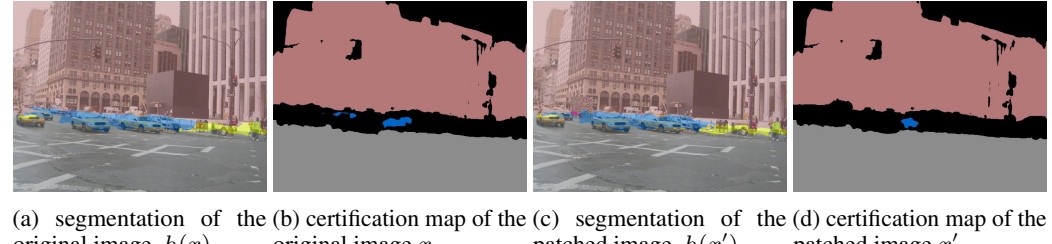

(a) segmentation of the original image, $h(x)$   (b) certification map of the original image $x$   (c) segmentation of the patched image, $h(x')$   (d) certification map of the patched image $x'$

Figure 12: Certified recovery for a 1 % patch used in the attack. The majority vote function $h$ recovers the prediction in presence of an adversarial patch that fools the undefended model. The segmentation for the original and patched image in (a) and (c) are the same for the regions certified in the certification maps (b) or (d). The certification maps (b) and (d) are also almost the same.

## E  ADDITIONAL EXPERIMENTS

In Tables 5 and 6, we provide additional experimental results for evaluating different masking schemes proposed in Section 4.1 on different models.

## F  INPAINTING ABLATION STUDIES

We perform ablation studies with respect to the demasking step. The results are in Table 7. Figure 14 provides additional illustrations. As can be seen from the results, our method heavily benefits from having available stronger inpainting models that allow achieving better clean and certified accuracy. We consider this property actually as a strength of our method since it will automatically benefit from future research and developments of stronger inpainting methods. For certified recovery, we also compare it to GIN Li et al. (2020) based on a generative model that we trained on ADE20K (without using style losses based on ImageNet trained VGG). The results are in Table 8. Illustrations can be found in Figure 15.

Table 5: The certified detection results(%) for a patch occupying no more than 1% of the image. mIoU - mean intersection over union, mR - mean recall, cmR - certified mean recall. %C - mean percentage of certified and correct pixels in the image.

| dataset | segm | mask | mIoU | big | | all | | %C |
|---|---|---|---|---|---|---|---|---|
| | | | | mR | cmR | mR | cmR | |
| ADE20K | BEiT-L | col | 56.33 | 74.26 | 61.15 | 68.40 | 35.88 | 65.44 |
| | | row | | | 52.77 | | 30.25 | 60.48 |
| | PSPNet | col | 44.39 | 61.83 | **50.02** | 54.74 | **26.37** | **60.57** |
| | | row | | | 42.44 | | 19.88 | 54.62 |
| | Swin | col | 48.13 | 68.51 | **55.45** | 59.13 | **29.06** | **61.44** |
| | | row | | | 47.21 | | 22.04 | 55.93 |
| COCO10K | PSPNet | col | 37.76 | 71.71 | **56.86** | 49.65 | **26.80** | **47.61** |
| | | row | | | 51.05 | | 23.51 | 43.40 |
| | DeepLab v3 | col | 37.81 | 72.52 | **56.54** | 49.98 | **26.86** | **47.17** |
| | | row | | | 50.51 | | 23.89 | 43.19 |

Table 6: The certified recovery results(%) against a 0.5% patch. 3-mask and 4-mask correspond to $T = 3$ and $T = 4$ respectively (Figure 2)

| dataset | segm | mask | mIoU | big | | all | | %C |
|---|---|---|---|---|---|---|---|---|
| | | | | mR | cmR | mR | cmR | |
| ADE20K | BEiT-L | col | 28.64 | 71.95 | 50.84 | 34.65 | 16.04 | 47.76 |
| | | row | 18.82 | 53.77 | 21.24 | 22.74 | 5.95 | 32.30 |
| | | 3-mask | 22.40 | 64.83 | 33.96 | 26.89 | 8.97 | 39.59 |
| | | 4-mask | 19.93 | 60.90 | 25.03 | 24.22 | 6.43 | 35.01 |
| | PSPNet | col | **19.17** | **51.90** | **34.11** | **23.66** | **10.76** | **44.90** |
| | | row | 12.00 | 36.26 | 12.03 | 15.03 | 3.74 | 28.29 |
| | | 3-mask | 15.00 | 44.93 | 19.55 | 18.41 | 5.58 | 35.85 |
| | | 4-mask | 12.74 | 40.41 | 15.86 | 15.87 | 4.14 | 31.22 |
| | Swin | col | **22.43** | **59.75** | **34.88** | **27.09** | **11.70** | **46.14** |
| | | row | 13.58 | 42.88 | 15.13 | 16.70 | 4.46 | 30.64 |
| | | 3-mask | 17.06 | 51.03 | 24.15 | 20.74 | 6.65 | 38.27 |
| | | 4-mask | 14.77 | 46.67 | 17.74 | 18.05 | 4.72 | 34.04 |
| COCO10K | PSPNet | col | **21.94** | **61.56** | **36.67** | **29.94** | **11.13** | **29.51** |
| | | row | 18.87 | 58.04 | 20.90 | 26.16 | 6.14 | 19.31 |
| | | 3-mask | 18.82 | 59.26 | 29.00 | 25.85 | 7.56 | 25.21 |
| | | 4-mask | 17.46 | 58.47 | 23.63 | 24.35 | 5.51 | 20.36 |
| | DeepLab v3 | col | **23.12** | **62.60** | **33.84** | **31.59** | **11.55** | **28.71** |
| | | row | 20.04 | 55.71 | 17.80 | 27.89 | 6.28 | 17.04 |
| | | 3-mask | 20.14 | 58.02 | 27.14 | 27.82 | 8.05 | 24.30 |
| | | 4-mask | 19.35 | 58.22 | 22.01 | 26.74 | 5.79 | 19.38 |

Table 7: Comparison for demasked smoothing with and without demasking step. mIoU - mean intersection over union, mR - mean recall, cmR - certified mean recall. %C - mean percentage of certified and correct pixels in the image. We use Swin model on 200 ADE20K images with column masking for certified detection and certified recovery. We compare masking the columns with solid black color without demasking to ZITS demasking.

| mode | patch size | demasking | mIoU | big | | all | | %C |
|------|------------|-----------|------|-----|-----|-----|-----|-----|
| | | | | mR | cmR | mR | cmR | |
| detection | 1.0% | ✓ | 38.56 | 67.25 | **58.85** | 53.37 | **23.35** | **62.89** |
| | | ✗ | | | 19.49 | | 3.09 | 21.19 |
| recovery | 0.5% | ✓ | **19.09** | **66.03** | **52.71** | **23.02** | **12.66** | **47.05** |
| | | ✗ | 1.10 | 15.09 | 7.71 | 1.79 | 0.72 | 18.59 |

Table 8: Comparison of our two demasking methods: ZITS and GIN. mIoU - mean intersection over union, mR - mean recall, cmR - certified mean recall. %C - mean percentage of certified and correct pixels in the image. We use Swin model on 200 ADE20K images.

| demasking | trained on | mIoU | big | | all | | %C |
|-----------|------------|------|-----|-----|-----|-----|-----|
| | | | mR | cmR | mR | cmR | |
| ZITS (Dong et al., 2022) | Places2 | **19.09** | **66.03** | **52.71** | **23.02** | **12.66** | **47.05** |
| GIN (Li et al., 2020) | ADE20K | 5.46 | 32.27 | 19.05 | 7.62 | 3.52 | 32.08 |

Table 9: Comparison of different inpainting models: LAMA Suvorov et al. (2022) and ZITS Dong et al. (2022). mIoU - mean intersection over union, mR - mean recall, cmR - certified mean recall. %C - mean percentage of certified and correct pixels in the image. For detection, we provide clean mIoU since the output is unaffected and mean false alert rate (FAR) (lower is better).

| mode | mask | demasker | mIoU | big | | all | | %C | FAR ↓ |
|------|------|----------|------|-----|-----|-----|-----|-----|-------|
| | | | | mR | cmR | mR | cmR | | |
| detection | column | ZITS | 53.08 | 70.92 | **57.33** | 64.45 | **32.55** | **63.55** | **20.04** |
| | | LAMA | | | 56.99 | | 31.67 | 64.21 | 19.37 |
| 1% patch | row | ZITS | 53.08 | 70.92 | 50.05 | 64.45 | 26.65 | 58.34 | 25.24 |
| | | LAMA | | | 49.06 | | 26.58 | 59.21 | 24.38 |
| | column | ZITS | **24.92** | **60.77** | **41.26** | **29.84** | **12.98** | **46.22** | |
| | | LAMA | 22.48 | 58.20 | 37.51 | 26.49 | 11.49 | 45.95 | |
| recovery | row | ZITS | 16.33 | 46.91 | 16.72 | 19.51 | 4.83 | 31.71 | N/A |
| | | LAMA | 15.64 | 43.07 | 16.51 | 18.78 | 4.95 | 32.84 | |
| 0.5% patch | 3-mask | ZITS | 19.90 | 56.90 | 26.51 | 23.86 | 7.54 | 38.64 | |
| | | LAMA | 18.54 | 53.59 | 27.39 | 22.12 | 7.58 | 39.52 | |
| | 4-mask | ZITS | 18.82 | 52.96 | 23.75 | 22.56 | 5.87 | 34.36 | |
| | | LAMA | 17.00 | 50.60 | 18.18 | 20.22 | 5.23 | 35.98 | |

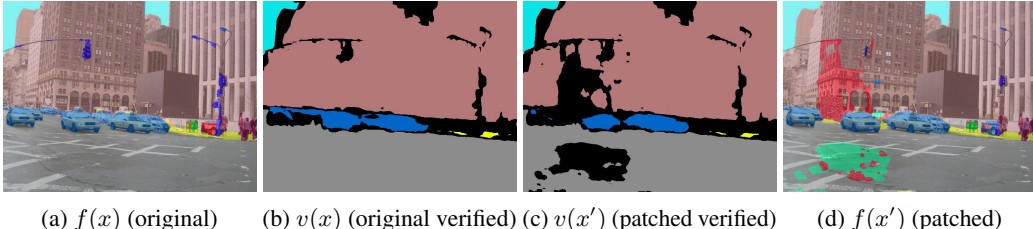

(a) $f(x)$ (original)    (b) $v(x)$ (original verified) (c) $v(x')$ (patched verified)    (d) $f(x')$ (patched)

Figure 13: $f$ is a segmentation model (Swin Liu et al. (2021)) and $v$ is the verification function (Section 4.2). For an attacked image $x'$ $v(x')$ detects the region of $f(x')$ which was (potentially) affected by an adversarial patch.

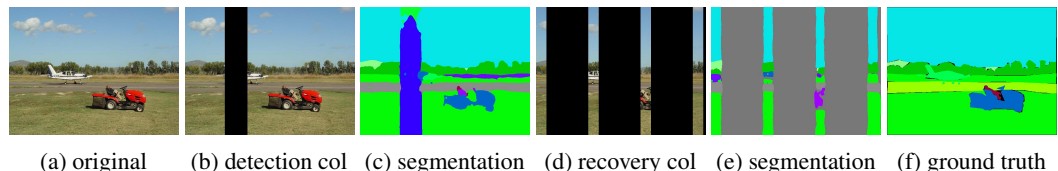

(a) original    (b) detection col  (c) segmentation  (d) recovery col  (e) segmentation  (f) ground truth

Figure 14: Results without image demasking. The solid color inpainting is treated as a separate object in the scene because we need to classify every pixel in semantic segmentation task. Therefore, it is hard to achieve a situation where all the demasked segmentation agree on some pixel which is represented in the Table 7.

## G    COMPARISON TO SIMPLIFIED DERANDOMIZED SMOOTHING

Derandomized Smoothing (DRS) Levine & Feizi (2020) was proposed for certified recovery, therefore in this section we focus on this task. Direct adaptation of derandomized smoothing to semantic segmentation task requires training a model that is able to predict the full image segmentation from a small visible region. Since it is not immediately clear to us what architectural design and training procedure would be needed to train such a model, we consider a simplified version of DRS that we call DRS-S. In this version, we consider an off-the-shelf semantic segmentation model and evaluate how it performs with column masking from DRS. Therefore, we do not encode the masked regions with the special 'NULL' value like in DRS but use black color instead. That is because an off-the-shelf model cannot work with 'NULL' values.

We run our experiments on ADE20K dataset. We consider the DRS parameters from the recent SOTA version of Derandomized Smoothing by Salman et al. Salman et al. (2021). They use column width $b = 19$ and stride $s = 10$ for certified classification of 224x224 ImageNet images. To account for the fact that ADE20K images have larger resolution than ImageNet, we scale the parameters to column width $b = 42$ and stride $s = 22$. To make the comparison consistent with the rest of our results, we use the patch occupying $0.5\%$ of the image.

From Table 10 we can see that DRS-S performs poorly on semantic segmentation task. The reason for that is illustrated in Figure 16. Processing the column region in 16c would probably be sufficient for a classification model to classify the image into the class "house". But it is clearly not sufficient to reconstruct the whole segmentation map 16e as can be seen in the Figure 16g. Whether doing this would be possible with a model specifically trained to reconstruct the segmentation map from a very small visible region is an open research question (up to our knowledge).

We point out that the value %C of certified and correctly classified pixels in the Table 10 is still surprisingly high for DRS-S compared to other metircs. We attribute this to the fact that the solid black regions are usually treated as a wall by the segmentation model, therefore the images are usually segmented as a wall by the DRS majority voting. And the wall is a common part of both indoor and outdoor scenes in ADE20K as can be implied from the Table 11 of "big" ADE20K classes. Therefore, always classifying the output as a wall provides a decent fraction of correctly classified pixels because of the skewed classes.

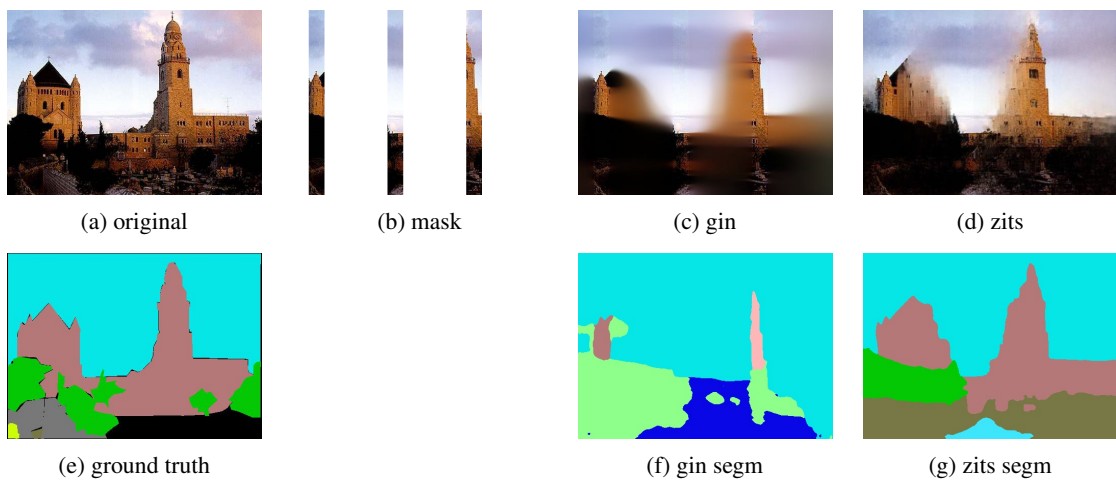

(a) original      (b) mask      (c) gin      (d) zits

(e) ground truth      (f) gin segm      (g) zits segm

Figure 15: Comparison between GIN and ZITS inpainting

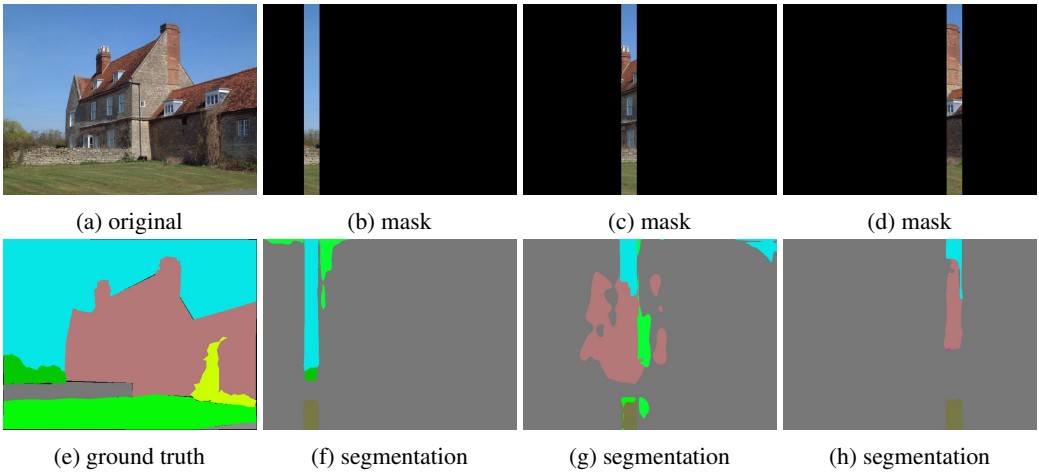

(a) original      (b) mask      (c) mask      (d) mask

(e) ground truth      (f) segmentation      (g) segmentation      (h) segmentation

Figure 16

However, to provide a better comparison with DRS, we emulate the model which is able to reconstruct the whole segmentation map from the column masking proposed in DRS. We do this by applying the demasking approach proposed in this work. We first try to reconstruct the whole image from one column and then segment it with an off-the-shelf model as we did with the masks proposed in this paper. We call this approach DRS-E and the results can be found in Table 10.

Table 10: Comparison of our method with simplified Derandomized Smoothing Levine & Feizi (2020). mIoU - mean intersection over union, mR - mean recall, cmR - certified mean recall. %C - mean percentage of certified and correct pixels in the image. We use Swin model on 200 ADE20K images.

| method | mIoU | big | | all | | %C |
|--------|------|-----|-----|-----|-----|-----|
| | | mR | cmR | mR | cmR | |
| Demasked (our) | **19.09** | **66.03** | **52.71** | **23.02** | **12.66** | **47.05** |
| DRS-S | 0.42 | 11.35 | 9.08 | 1.04 | 0.83 | 28.01 |
| DRS-E | 9.12 | 54.67 | 41.78 | 11.04 | 7.86 | 45.03 |

Table 11: The list of 19 "big" classes for ADE20K (Zhou et al., 2017) (out of 150 classes in total) with their average fraction of occupied pixels in the images where they are present (%) and index in the list of dataset classes. We define a class to be "big" if it occupies on average more than 20% of the pixels in the images in which this class appears.

| # | index | name | fraction | # | index | name | fraction |
|---|---|---|---|---|---|---|---|
| 1 | 0 | wall | 25.88 | 11 | 79 | hovel | 25.93 |
| 2 | 1 | building | 32.36 | 12 | 88 | booth | 23.91 |
| 3 | 2 | sky | 21.54 | 13 | 96 | escalator | 20.96 |
| 4 | 7 | bed | 21.25 | 14 | 103 | ship | 26.81 |
| 5 | 21 | water | 22.10 | 15 | 104 | fountain | 28.81 |
| 6 | 29 | field | 22.97 | 16 | 107 | washer | 22.07 |
| 7 | 46 | sand | 21.22 | 17 | 109 | swimming pool | 28.87 |
| 8 | 48 | skyscraper | 42.92 | 18 | 114 | tent | 34.57 |
| 9 | 54 | runway | 28.05 | 19 | 128 | lake | 34.57 |
| 10 | 55 | case | 37.57 | | | | |

Table 12: The list of 21 "big" classes for COCO-Stuff-10K (Caesar et al., 2018) (out of 171 classes in total) with their average fraction of occupied pixels in the images where they are present (%) and index in the list of dataset classes. We define a class to be "big" if it occupies on average more than 20% of the pixels in the images in which this class appears.

| # | index | name | fraction | # | index | name | fraction |
|---|---|---|---|---|---|---|---|
| 1 | 6 | bus | 21.46 | 11 | 105 | floor-stone | 20.10 |
| 2 | 7 | train | 23.11 | 12 | 111 | fruit | 20.48 |
| 3 | 20 | cow | 24.17 | 13 | 113 | grass | 23.25 |
| 4 | 21 | elephant | 28.50 | 14 | 134 | playingfield | 38.64 |
| 5 | 49 | sandwich | 23.99 | 15 | 137 | river | 40.01 |
| 6 | 51 | broccoli | 20.18 | 16 | 143 | sand | 26.37 |
| 7 | 54 | pizza | 25.86 | 17 | 144 | sea | 36.51 |
| 8 | 60 | bed | 36.86 | 18 | 146 | sky-other | 22.94 |
| 9 | 61 | dining table | 21.71 | 19 | 148 | snow | 51.60 |
| 10 | 95 | clouds | 24.11 | 20 | 159 | vegetable | 20.35 |
| | | | | 21 | 167 | water-other | 21.67 |

## H  A LIST OF BIG CLASSES

In Section 5.2 we suggest another perspective on the evaluation of our DEMASKED SMOOTHING by specifically considering its performance on "big" semantic classes. The object of these classes occupy on average more than 20% of the images in which they appear. Correctly segmenting these classes is important for understanding the scene. In Tables 11 and 12 we provide the full list of such classes in ADE20K (Zhou et al., 2017) and COCO-Stuff-10K (Caesar et al., 2018) respectively together with the average fraction of pixels that they occupy in the images in which they are present. We point out that for COCO-Stuff-10K some typically smaller classes such as "sandwich" or "fruit" get included in the list of big classes because of the macro-scale images in which they occupy a big part of the scene.

## I  COMPLEXITY ANALYSIS AND PARALLELIZATION

In DEMASKED SMOOTHING, we propose a set of $K$ masks that are applied to the original image (denote the cost of applying a single masking by $M$). As illustrated in Figure 1c, the masked images are demasked (denote the cost of demasking an image by $D$) and segmented (denote the cost of segmenting an image by $S$); thereupon per-mask segmentations are aggregated into a final segmentation and certification (cost of aggregation proportional to $K$). Asymptotically, compute

grows thus with $O(K(M + D + S) + K)$ while the cost of a standard segmentation is $O(S)$. Thus, for large $K$ or $M + D \gg S$, real-time applicability would actually be impractical. However, we note that:

1. $M + D$ is roughly of the same size as $S$ for typical DL-based inpainting and segmentation models.

2. For certified recovery, we operate in a setting where K is small ($K \in 5, 7, 9$) and does not grow with the image resolution. This is unlike Derandomized Smoothing and its derivatives, where the number of masks in the recovery task grows with the image resolution (or randomized smoothing with thousands of samples per input). This small value of K benefits our the method in time-sensitive applications. For certified detection, we can adjust the number of masks for the computational speed by using strided masking as suggested in Section 4.1.

3. Moreover, masking, demasking, and segmenting for different masks do not use any shared data and can thus be fully parallelized if sufficiently powerful hardware is available. Only the aggregation step requires the results of all the previous stages. However, aggregation time is small compared to the other stages. In terms of latency, a fully parallelized version of our procedure would thus have a latency proportional to $O(M + D + S + K)$. For small $K$ and $M + D \approx S$, application to real-time video can be facilitated by means of parallelization.

## J    USED DATA

In this work, we only use the datasets published under formal licenses: ADE20K (Zhou et al., 2017) and COCO-Stuff-10K (Caesar et al., 2018). To the best of our knowledge, data used in this project do not contain any personally identifiable information or offensive content. The models ZITS (Dong et al., 2022) and Swin (Liu et al., 2021) are published under Apache-2.0 license. The text of the license for PSPNet (Zhao et al., 2017) can be found here: `https://github.com/hszhao/PSPNet/blob/master/LICENSE`

## K    DEMASKED SMOOTHING VISUALIZATION

In this section, we provide additional illustrations of our method (Figures 17, 18, 19, 20). Similarly to the Table 1 we certify against a 1% patch for the detection task and against 0.5% patch for the recovery task. For each mask type we illustrate all the stages summarized in the Figure 1c. We also provide examples of certification maps for certified recovery and certified detection with different images (Figure 21, 22).

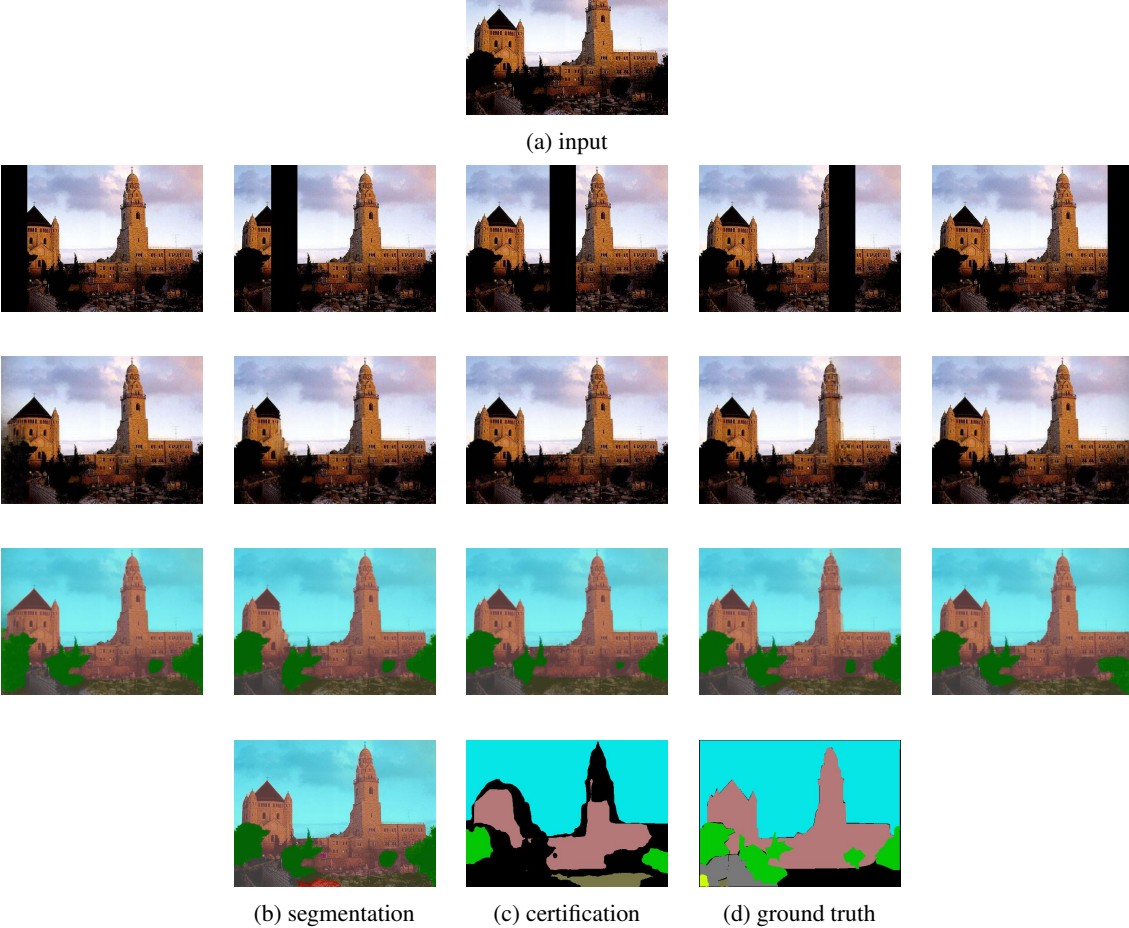

(a) input

(b) segmentation   (c) certification   (d) ground truth

Figure 17: DEMASKED SMOOTHING detection column masking illustration for an image from ADE20K (Zhou et al., 2017). We illustrate five masks out of twenty.

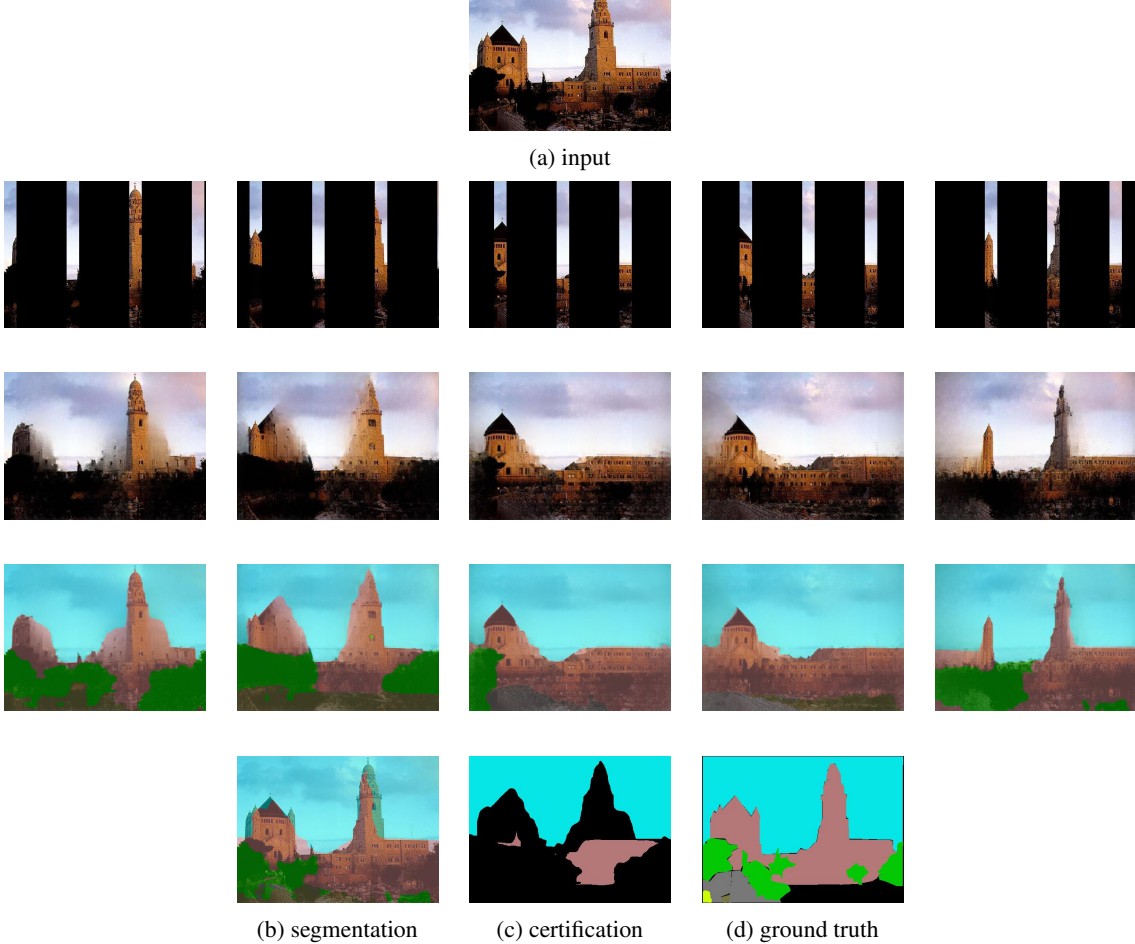

(a) input

(b) segmentation     (c) certification     (d) ground truth

Figure 18: DEMASKED SMOOTHING recovery column masking illustration for an image from ADE20K (Zhou et al., 2017).

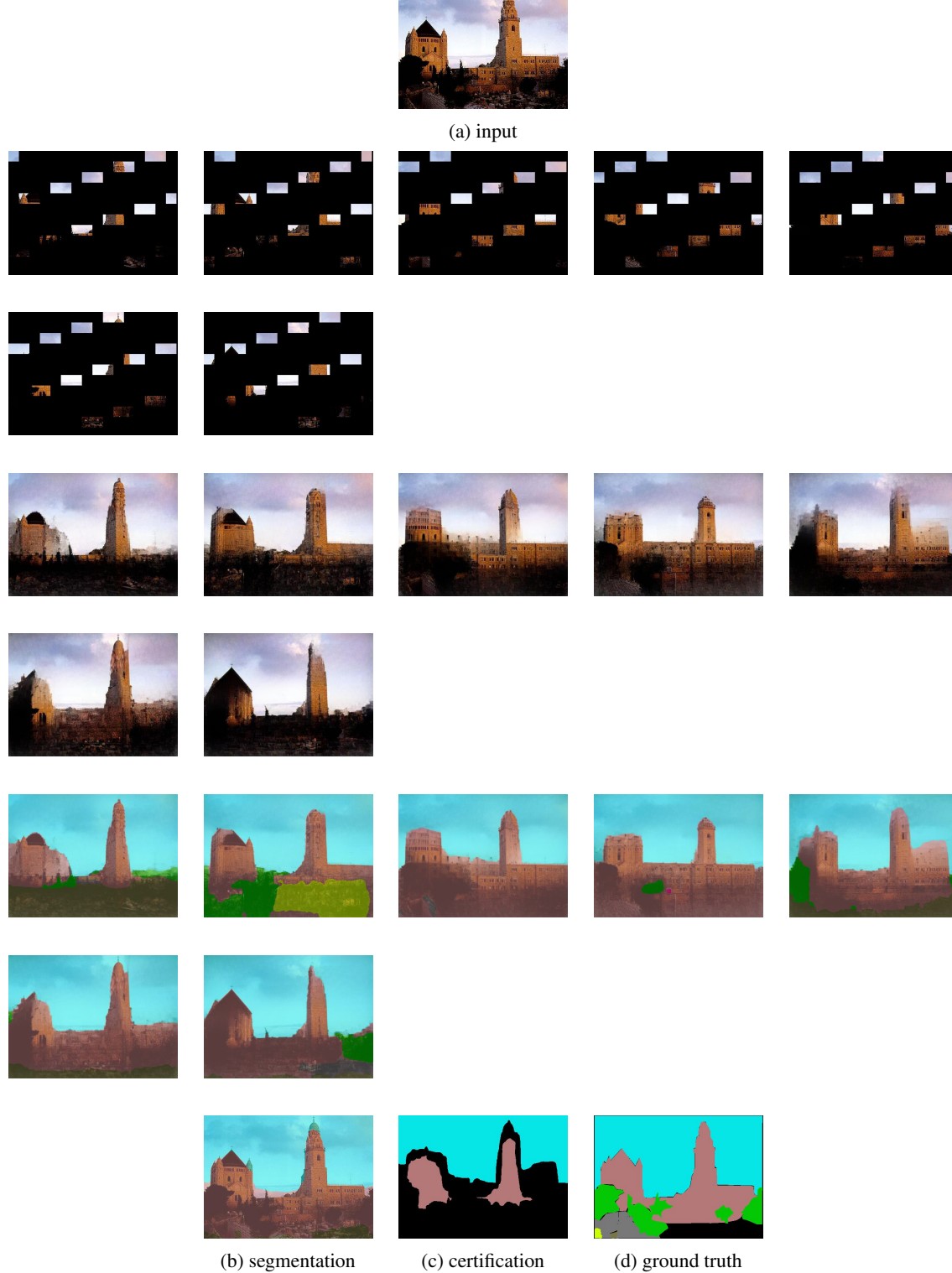

Figure 19: DEMASKED SMOOTHING recovery masking for $T = 3$, $K = 7$ masks (Section 4.1) illustration for an image from ADE20K.

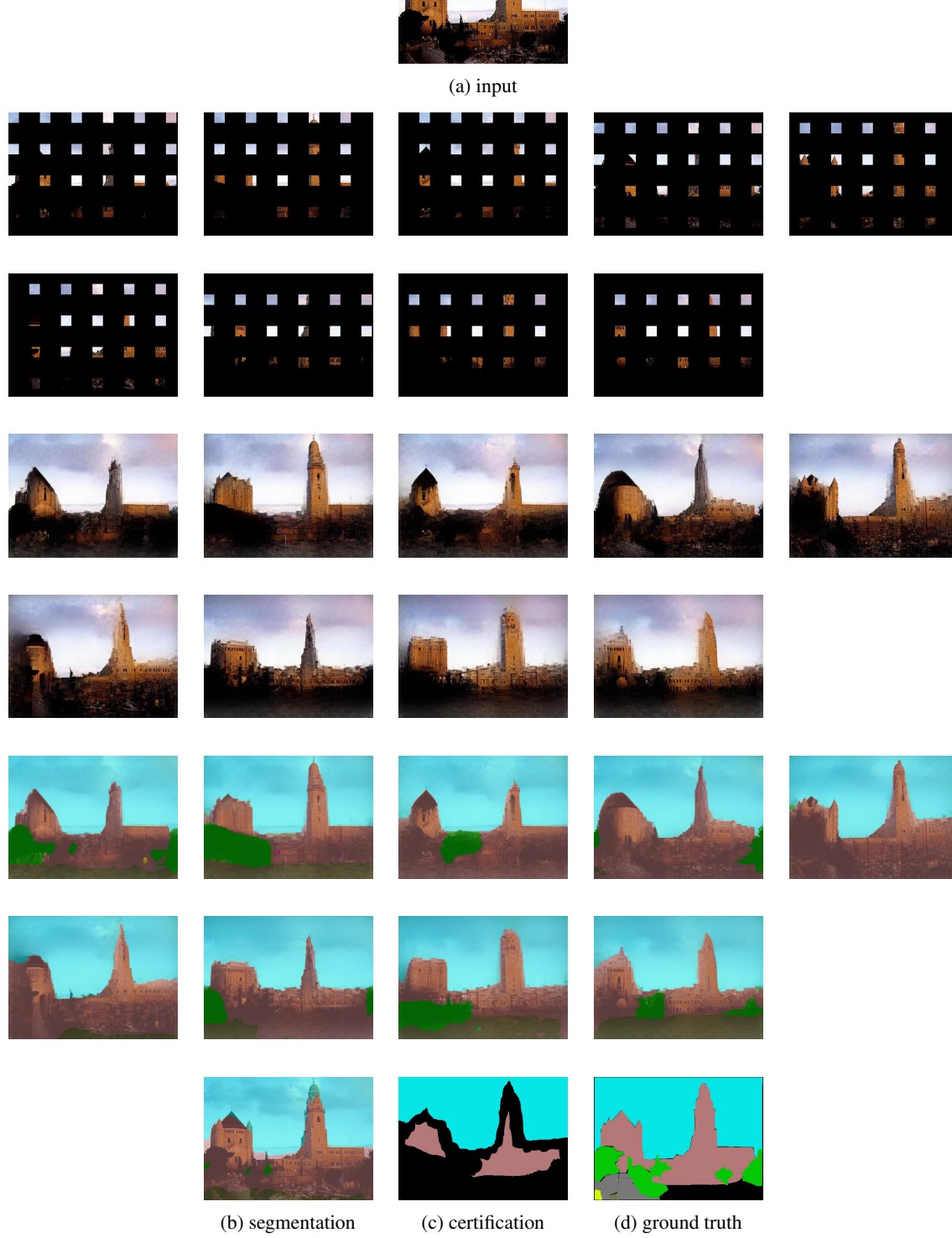

Figure 20: DEMASKED SMOOTHING recovery masking for $T = 4$, $K = 9$ masks (Section 4.1) illustration for an image from ADE20K (Zhou et al., 2017).

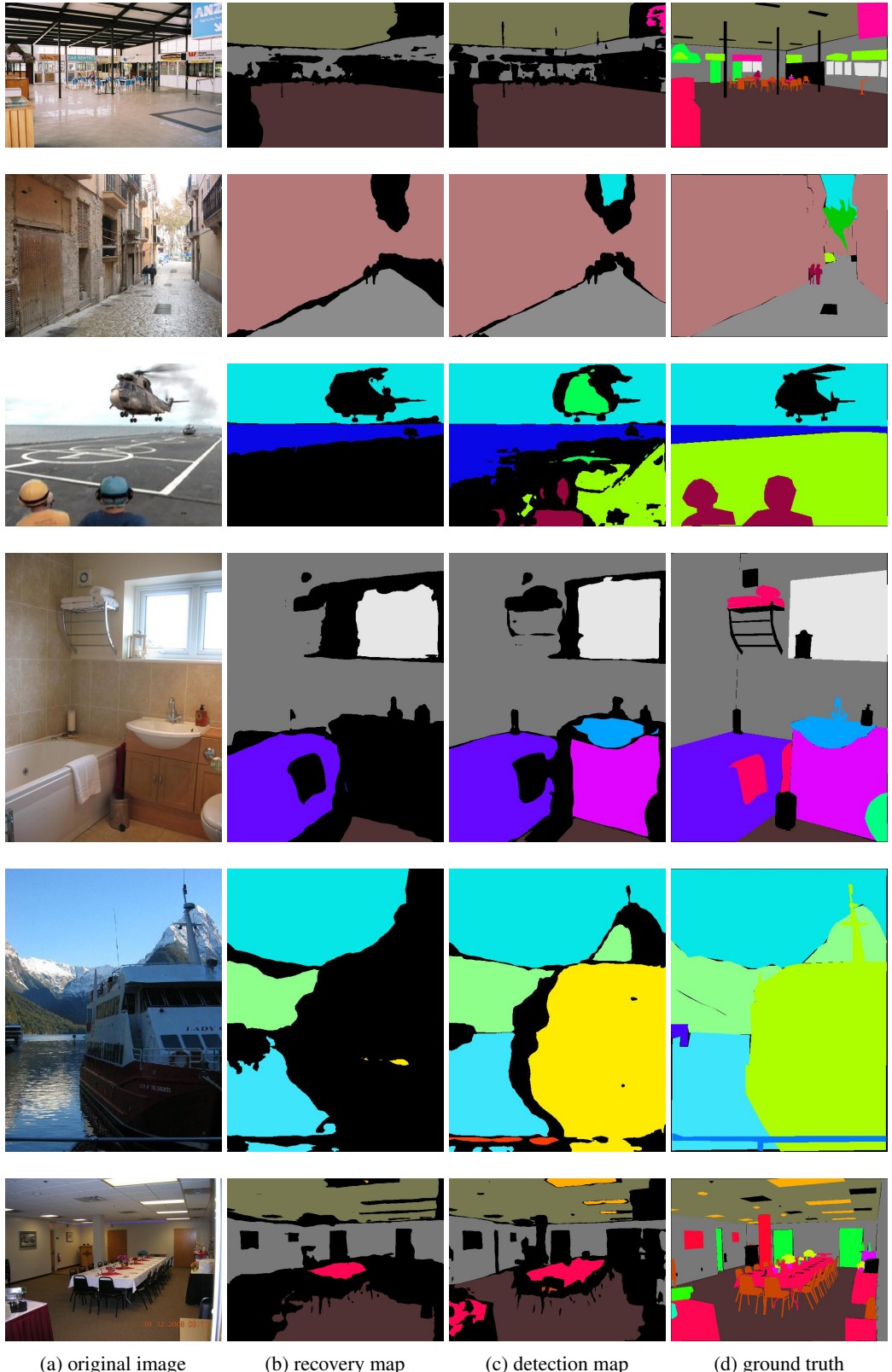

    (a) original image        (b) recovery map        (c) detection map        (d) ground truth

Figure 21: Certification map examples on ADE20K (Zhou et al., 2017) with ZITS (Dong et al., 2022) and Swin Liu et al. (2021).

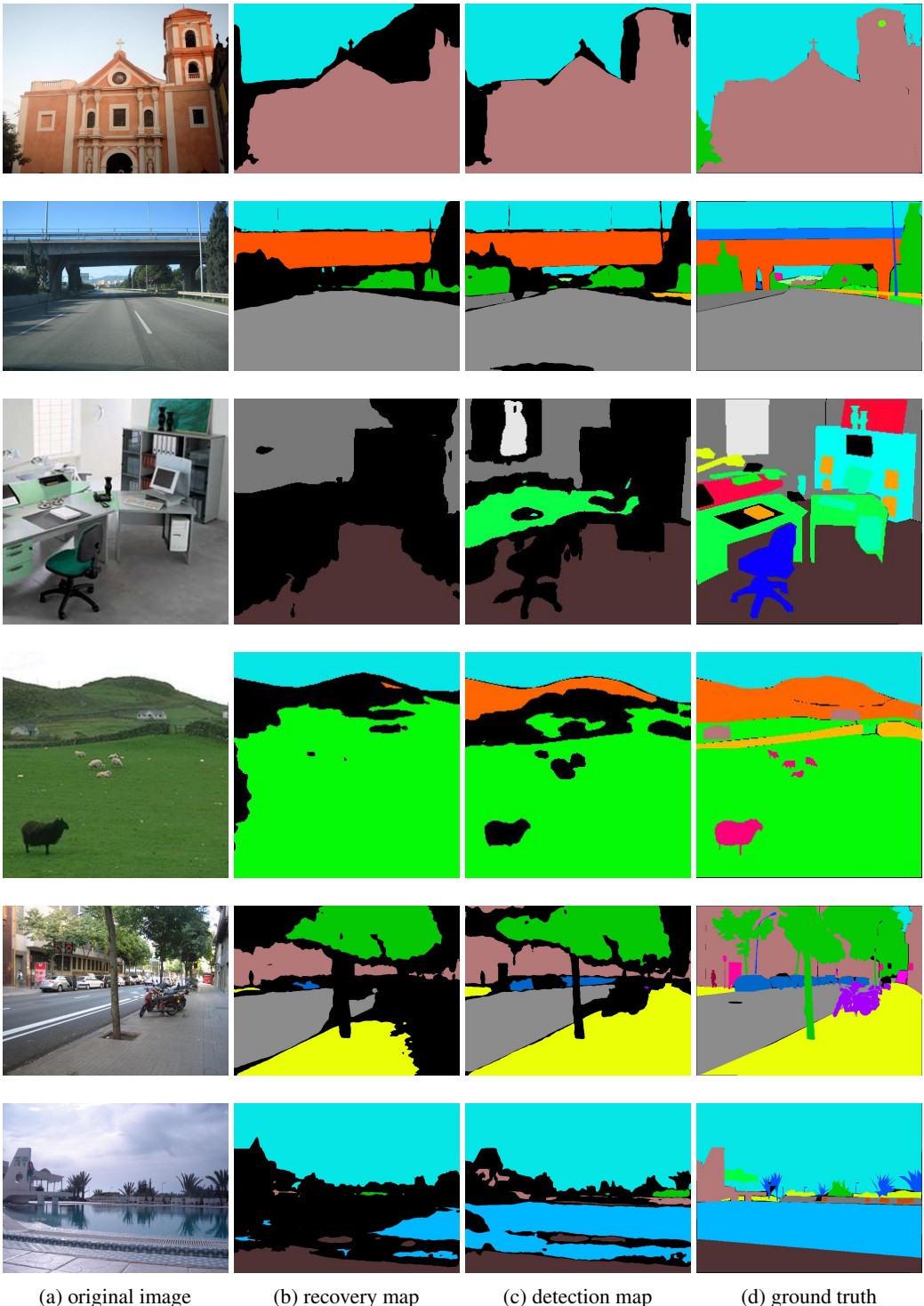

(a) original image     (b) recovery map     (c) detection map     (d) ground truth

Figure 22: Certification map examples on ADE20K (Zhou et al., 2017) with ZITS (Dong et al., 2022) and Swin Liu et al. (2021).

