# OpenReview forum: "Certified Defences Against Adversarial Patch Attacks on Semantic Segmentation"
_ICLR.cc/2023/Conference — ICLR 2023 poster_

### Official Review · Reviewer_F6Ro · 2022-10-16

**Confidence:** 3
**Correctness:** 4
**Technical Novelty And Significance:** 3
**Empirical Novelty And Significance:** 3
**Recommendation:** 8

**Clarity, Quality, Novelty And Reproducibility:**

As detailed above, the work is clear to follow. It seems reasonably reproducible based on the details in Algorithm 1 and Section 4. The authors detail the novelty compared to prior work in Section 2, describing the uniqueness.

**Details Of Ethics Concerns:**

None so far, authors addressed this in Section 7.

**Strength And Weaknesses:**

Strength:
* The work is well organized, clear to follow with no major grammatical errors
* The empirical evaluation seems reasonable, along with the prior work.
* The results on the benchmarked datasets seem promising.


Weaknesses:
* The theoretical grounding is not clearly explained in the main paper.
* Limitations are not clearly described, evaluation limited to semantic segmentation domain


**Summary Of The Paper:**

The work provides their "DEMASKED SMOOTHING" approach, which the authors argue provide defence framework against patch attacks for segmentation models. They evaluate their model on semantic segmentation datasets ADE20k and COCO-Stuff-10K. They use models trained on existing benchmarks, evaluated on various masking schemes. The authors discuss the results and approach compared to baselines in Section 5.

**Summary Of The Review:**

Overall the work is well organized, described clearly and makes an original contribution of reasonable significance. The empirical results look promising, although it seems authors skipped some detailed on limitations and expanding theoretical claims.

---

> ### Author Response · Authors · 2022-11-17
> **Addressing the questions**
>
> We thank the reviewer for the provided comments. We address the questions raised in the section "Weaknesses".
>
> 1. "The theoretical grounding is not clearly explained in the main paper."
>
> We have made the proofs of Theorem 1 and Theorem 2 more detailed. We have also added Figure 7 and Figure 8 to schematically illustrate the ideas of the proofs.
>
> 2. "Limitations are not clearly described, evaluation limited to semantic segmentation domain"
>
> We have added the discussion on the limitations of our approach in the Section 5.4 of the manuscript.
>
> The performance of Demasked Smoothing certified recovery may be insufficient for the downstream task if we certify against big patches (Figure 5) unless robustness is prioritized over clean performance. We point out that robustly segmenting small objects is fundamentally difficult under the adversarial patch threat model since the objects themselves can be completely or partially covered by an adversarial patch which makes it impossible to properly segment them even for a human being. Demasked Smoothing certification requires an upper bound on the size of the expected patch (Section 4).

---

### Official Review · Reviewer_TG16 · 2022-10-24

**Confidence:** 4
**Clarity, Quality, Novelty And Reproducibility:** 1. While it is true that the defense …
**Correctness:** 3
**Technical Novelty And Significance:** 3
**Empirical Novelty And Significance:** 3
**Recommendation:** 6

**Strength And Weaknesses:**

While to my knowledge I agree that this paper is one of the first certified defenses proposed for semantic segmentation, I have several questions about the current presentation of the results (see the following sections).

**Summary Of The Paper:**

This paper proposed Demasked Smoothing (DS), a certified defense against adversarial patch attacks on semantic segmentation. The empirical performance outruns baseline certified defenses such as DRS.

**Summary Of The Review:**

Sufficient novely on certified defense for semantic segmentation, though the certification seems rather restrictive (e.g., requiring very high consistency of image recovery rather than majority voting). More discussions on the working mechanism, limitations, and additional comparisons are needed.

---

> ### Author Response · Authors · 2022-11-17
> **Response and additional experiments (part 2)**
>
> 2. "The certification criterion seems to be largely different from certified patch defense, in that there is no notion of majority vote on masked parts. Rather, the proposed defense aims at recovering a consistent pattern of the masked part and uses it for the final certification. In a way, the recovery is either correct or incorrect, and there is no notion of correctness when there is some inconsistency of demasked results, which intuitively should give stronger certified results. Can the authors comment on this? Also, if I understand correctly, the proposed method cannot certify the maximum size of a certifiable patch, which is another limitation. The certification is more similar to robust segmentation recovery using multiple damasked segmentations from an inpainter. I hope the authors can comment more on the limitation of the proposed method."
>
> In certified recovery, we consider a pixel to be certified if we can guarantee that on an input with an adversarial patch our method will make the same prediction for this pixel as on the clean input. In Demasked Smoothing, we require the masked predictions on a pixel to be univocal to certify this pixel. One alternative could be to consider a margin between the most predicted class and a runner-up as it was, for example, done in Derandomized smoothing [1]. However, it would require us to have significantly more masks for each image. Increasing the number of masks is not desirable for semantic segmentation task because, unlike image classification, each mask needs to keep a significant part of the image visible to allow understanding the scene. Increasing the number of masks would lead to decreasing the region visible in each of these masks which would have a detrimental effect on the certification performance.
>
> Similarly to many other certified defences against patches [1], our method requires an upper bound on the patch size to perform certification. In the paper we provide the effect of the patch size on the method performance (Figure 5). Increasing the patch size makes it harder for our method to certify the pixels, but even for relatively large patches (e. g. 5% patch for certified detection) our method provides non-trivial certification results. Thus, it is not entirely clear to us what is meant by "maximum size of certifiable patch" since we certify the image pixel-wise and for different pixels the patch size that we can certify against would be different.
>
> We have added the discussion on the limitations of our approach in the Section 5.4 of the manuscript. The performance of Demasked Smoothing certified recovery may be insufficient for the downstream task if we certify against big patches (Figure 5) unless robustness is prioritized over clean performance. We point out that robustly segmenting small objects is fundamentally difficult under the adversarial patch threat model since the objects themselves can be completely or partially covered by an adversarial patch which makes it impossible to properly segment them even for a human being. Demasked Smoothing certification requires an upper bound on the size of the expected patch (Section 4).
>
> [1] Alexander Levine and Soheil Feizi, "(De)Randomized Smoothing for Certifiable Defense against Patch Attacks", NeurIPS 2020
>
> 3. "To make a fair comparison to DRS and Randomized Cropping, can the authors compare semantic segmentation similar to their settings (e.g., using segmentation models robust to random masking)? Is there still any notable performance gain?"
>
> We are not aware of available segmentation models that are robust to random masking. Training such a model from scratch would require designing a specific architecture and training procedure which are beyond the scope of this work. This is the reason why we considered a simplified setting for comparison with Derandomized Smoothing in Table 2 in the paper.
>
> In this work, we consider deterministic certified defences. Randomized Cropping [1] is a non-deterministic defence i. e. the certification results only hold for a given confidence interval. Thus, a direct comparison with our work is infeasible.
>
> [1] Wan-Yi Lin, Fatemeh Sheikholeslami, Jinghao Shi, Leslie Rice, and J Zico Kolter. Certified robustness against physically-realizable patch attack via randomized cropping, 2021. URL https://openreview.net/forum?id=vttv9ADGuWF.

---

> ### Author Response · Authors · 2022-11-17
> **Response and additional experiments (part 1)**
>
> We highly appreciate the time devoted to reviewing our paper. We address the questions raised in the review.
>
> 1. "While it is true that the defense applies to any segmentation model, the defense requires the use of an off-the-shelf image inpainter, which could be the bottleneck (or even harmful) to certified performance if the inpainter is not reliable. Can the authors do an ablation study on the effect of image inpainter?"
>
> We have done additional ablation studies on the effect of the image inpainter. We have used the LAMA [1] inpainting approach based on Fast Fourier convolutional neural network. We have compared it to using ZITS [2] method based on incremental transformer structure. ZITS is reported to outperform LAMA on different inpainting metrics [2].
>
> In the Table C and Table D, we provide a detailed comparison with respect to all certification modes and masking schemes proposed in our work for the ADE20K [3] validation set and BEiT-B [4] segmentation model. We observe that using a different inpainting method has a marginal effect on the results of Demasked Smoothing certification demonstrating the reliability of the method.
>
> Table C. Certified recovery on ADE20K validation set (2000 images) for BEiT-B segmentation model against a 0.5% patch. mIoU - mean intersection over union, mR - mean recall, mR big - mean recall for "big" classes, cmR - certified mean recall, cmR big - certified mean recall for "big" classes. %C - mean percentage of certified and correct pixels in the image.
>
> | mask		    | inpainter	| mIoU (clean) | mR big (clean)     | cmR big     | mR all (clean)     | cmR all     | %C        |
> |:--------------|:--------------|:--------------|:-------------|:-------------------|:------------|:-------------------|:------------|
> | column    | ZITS		  | **24.92**        | **60.77**              | **41.26**   | **29.84**              | **12.98**   | **46.22**  |
> | column    | LAMA     		  | 22.48        | 58.20              | 37.51   | 26.49              | 11.49   | 45.95  |
> |
> | row | ZITS		  | **16.33**        | **46.91**              | **16.72**   | **19.51**              | 4.83   | 31.71  |
> | row | LAMA              | 15.64        | 43.07              | 16.51   | 18.78              | **4.95**   | **32.84** |
> |
> | 3-mask | ZITS		  | **19.90**        | **56.90**              | 26.51   | **23.86**              | 7.54   | 38.64 |
> | 3-mask | LAMA              | 18.54        | 53.59              | **27.39**   | 22.12              | **7.58**   | **39.52** |
> |
> | 4-mask | ZITS		  | **18.82**        | **52.96**              | **23.75**   | **22.56**              | **5.87**   | 34.36 |
> | 4-mask | LAMA              | 17.00        | 50.60              | 18.18   | 20.22              | 5.23   | **35.98** |
>
> Table D. Certified detection on ADE20K validation set (2000 images) for BEiT-B segmentation model against a 1% patch. mIoU - mean intersection over union, mR - mean recall, mR big - mean recall for "big" classes, cmR - certified mean recall, cmR big - certified mean recall for "big" classes. %C - mean percentage of certified and correct pixels in the image. FAR - mean false alert rate (lower is better).
>
> | mask		    | inpainter	| mIoU (clean) | mR big (clean)     | cmR big     | mR all (clean)     | cmR all     | %C        | FAR $\downarrow$ |
> |:--------------|:--------------|:--------------|:-------------|:-------------------|:------------|:-------------------|:------------|:----------|
> | column    | ZITS		  | 53.08        | 70.92              | **57.33**   | 64.45              | **32.55**   | 63.55 | 20.04 |
> | column    | LAMA     		  | 53.08   		| 70.92        | 56.99              | 64.45       | 31.67              | **64.21**       | **19.37**     |
> |
> | row | ZITS	| 53.08        | 70.92              | **50.05**   | 64.45              | **26.65**   | 58.34 | 25.24 |
> | row | LAMA    | 53.08        | 70.92              | 49.06       | 64.45              | 26.58       | **59.21**   | **24.38** |
>
>
> [1] Roman Suvorov, Elizaveta Logacheva, Anton Mashikhin, Anastasia Remizova, Arsenii Ashukha, Aleksei Silvestrov, Naejin Kong, Harshith Goka, Kiwoong Park, and Victor S. Lempitsky. Resolution-robust large mask inpainting with fourier convolutions. In WACV 2022.
>
> [2] Qiaole Dong, Chenjie Cao, and Yanwei Fu. Incremental transformer structure enhanced image inpainting with masking positional encoding. In CVPR 2022.
>
> [3] Bolei Zhou, Hang Zhao, Xavier Puig, Sanja Fidler, Adela Barriuso, and Antonio Torralba. Scene parsing through ade20k dataset. In CVPR 2017.
>
> [4] Hangbo Bao, Li Dong, Songhao Piao, and Furu Wei. BEit: BERT pre-training of image transformers. In ICLR 2022.

---

> ### Comment · Reviewer_TG16 · 2022-11-18
> **Thank you for your response**
>
> I thank the authors for preparing the response. It has addressed my concerns. I am also happy to see a discussion of the limitation included in the updated version. I've increased my rating score.

---

> > ### Author Response · Authors · 2022-11-23
> > **Thank you**
> >
> > Dear Reviewer,
> >
> > We are glad that our response has addressed your concerns. We would like to thank you again for appreciating our work and recognizing our contributions!
> >
> > Best,
> >
> > The Authors

---

### Official Review · Reviewer_FoVC · 2022-10-24

**Confidence:** 3
**Correctness:** 2
**Technical Novelty And Significance:** 2
**Empirical Novelty And Significance:** 3
**Recommendation:** 6

**Clarity, Quality, Novelty And Reproducibility:**

The manuscript is hard to follow (eg. paragraph certified recovery in section 4.1) and has several weaknesses (see above).

**Strength And Weaknesses:**

Strengths
1. adversarial robustness is an important problem
2. a previous image-wide approach [salman21arxiv] has been extended for the case of dense prediction.

Weaknesses
1. This approach is likely going to strongly diminish semantic accuracy at small objects.
2. The majority of input pixels is inpainted. The manuscript does not seem to consider opportunities of attacking the inpainting algorithm either through adversarial patches or through data poisoning.
3. The second to last sentence of the proof to theorem 1 is not convincing.
4. Unclear implied guarantees of certification. Consider the example from Figure 18. An adversarial patch has opportunity to affect only T=4 of K=9 masks. Still, the model is going to make mistakes in the remaining K-T=5 masks. Could these mistakes lead to adversarial patch pass unnoticed?
5. Unclear relation with respect to adversarial training. Would it make sense to combine these two approaches?


**Summary Of The Paper:**

The manuscript addresses adversarial patch attacks for segmentation models. The proposed approach extends previous work on certified patch robustness for image-wide models. It is capable both of certified detection and certified recovery and it can be applied to any semantic segmentation model. The main idea is to apply a set of K masks to the input image, inpaint the void areas with ZITS, and then to detect or undo adversarial patch attack by analyzing all predictions at the given pixel.


**Summary Of The Review:**

The reviewer perceives moderate novelty, tough reading and several weaknesses.

---

> ### Author Response · Authors · 2022-11-17
> **Response**
>
> We thank the reviewer for the the time devoted to reviewing our paper. We address the questions raised in the section "Weaknesses".
>
> 1. "This approach is likely going to strongly diminish semantic accuracy at small objects."
>
> We point out that robustly segmenting small objects is fundamentally difficult under the employed adversarial patch threat model since the objects themselves can be completely or partially covered by an adversarial patch which makes it impossible to properly segment them even for a human being.
>
> We only observe a drop in semantic segmentation accuracy of Demasked Smoothing (in particular, for segmenting small objects) for the certified recovery mode (Tables 1, 3). Certified recovery provides a stronger robustness guarantee that comes at a cost of clean accuracy as usually happens in certified robustness field. When segmenting small object is crucial for the application, we recommend using certified detection approach proposed in this work that provides a weaker robustness guarantee but allows to maintain the standard segmentation accuracy of the model (although false alarms may be issued, see Table 1, 3).
>
> 2. "The majority of input pixels is inpainted. The manuscript does not seem to consider opportunities of attacking the inpainting algorithm either through adversarial patches or through data poisoning. "
>
> If some pixels of the adversarial patch are not masked, then the patch can potentially affect both a segmentation model and an inpainting model used by Demasked Smoothing. However, when the patch is completely masked, there is no way for it to affect any component of the further processing, in particular, the inpainting model. By design of our masking schemes we ensure that the number of maskings not affected by a patch is high enough to allow the recovery of the original prediction or detecting that it was affected. Thus optimizing the patch to affect the inpainting model or using patch-based data poisoning will not break the robustness guarantee. Other types of data poisoning (not using patches) are beyond the scope of this work.
>
> 3. "The second to last sentence of the proof to theorem 1 is not convincing. "
>
> We have made the proofs of Theorem 1 and Theorem 2 in the Appendix A of the manuscript more detailed. We have also added Figure 7 and Figure 8 to schematically illustrate the ideas of the proofs.
>
> 4. "Unclear implied guarantees of certification. Consider the example from Figure 18. An adversarial patch has opportunity to affect only T=4 of K=9 masks. Still, the model is going to make mistakes in the remaining K-T=5 masks. Could these mistakes lead to adversarial patch pass unnoticed?"
>
> Certified recovery guarantees that for certified pixels the prediction on the attacked image $x^\prime$ will be the same as for the clean image $x$. In Figure 18, the pixels for which all $K=9$ masks agree in $x$ are certified. If $T=4$ masks are affected by a patch in $x^\prime$, the prediction on the rest $K-T=5$ masks will be the same as for $x$. Since $K-T > T$, the majority voting result on $x^\prime$ will be the same as on $x$.
>
> We emphasize that the original prediction on $x$ can be incorrect with respect to the ground truth $y$. We do not guarantee that certified pixels will also be classified *correctly*. We only guarantee that the classification result will be the same on the clean and any adversarial image in our threat model.
>
> 5. "Unclear relation with respect to adversarial training. Would it make sense to combine these two approaches?"
>
> Adversarial training depends on the adversarial attack used to create adversarial examples during training. It does not provide guarantees against unknown adversarial attacks.
>
> When the patch is completely masked, the patch has no way to affect the prediction, no matter which optimization method was used to obtain it. However, if adversarial training would (as a byproduct) increase robustness of the model to suboptimal inpaintings, our procedure would benefit from it. But we note that direct training of the segmentation model on images with masking and inpainting would be more likely to increase performance. However, training the models specifically for certified robustness is beyond the scope of this work since we focus on the methods that require no additional training.

---

> > ### Comment · Reviewer_FoVC · 2022-11-22
> > **Thanks for the discussion**
> >
> > I am not an expert in adversarial training, but it seems that adversarial training with PGD offers a good overall performance.
> >
> > I thank the authors for their discussion. I have increased my rating to 6 - marginally above the acceptance threshold.

---

> > > ### Author Response · Authors · 2022-11-23
> > > **Thank you**
> > >
> > > Dear Reviewer,
> > >
> > > We are happy that our response has answered your questions on our manuscript. We thank you again for appreciating our work and recognizing our contributions!
> > >
> > > Best,
> > >
> > > The Authors

---

### Official Review · Reviewer_XUU4 · 2022-10-27

**Confidence:** 4
**Correctness:** 4
**Technical Novelty And Significance:** 3
**Empirical Novelty And Significance:** 3
**Recommendation:** 6

**Clarity, Quality, Novelty And Reproducibility:**

The clarify and quality are good.
For the reproducibility, it is reasonable to me. However, if the author can release the code, it would be much better.

**Strength And Weaknesses:**

Overall, the paper is interesting and extend the certification for semantic segmentation.

Strength:
1) This paper is well-written.
2) This is the first method to certifiably defend against adversarial patch attack on semantic segmentation.
3) This method can be applied on any off-the-shelf segmentation model without finetuning or any other adaptation.

Weakness:
1. One minor weakness is that this paper did not apply to the STOA transformer based methods such as Segformer. Hope to see the performance on Segformer.
2. Hope the author can provide the code to check the reproducibility of this paper.

[1] SegFormer: Simple and Efficient Design for Semantic Segmentation with Transformers.

**Summary Of The Paper:**

This paper presents a new method to certify the robustness of semantic segmentation models against adversarial patch attacks without extra training.


**Summary Of The Review:**

It implements certified defense under patch attacks on semantic segmentation.

---

> ### Author Response · Authors · 2022-11-17
> **Response**
>
> We are grateful for the provided review of our work. We address the questions raised in the section "Weaknesses".
>
> 1. "One minor weakness is that this paper did not apply to the STOA transformer based methods such as Segformer. Hope to see the performance on Segformer. "
>
> We provide additional experimental results for the BEiT-L [1] segmentation model which achieves 56.33 mIoU on the ADE20K validation set. It significantly outperforms the best performing Segformer model reported to have 51.13 mIoU. Plugging the BEiT-L model into Demasked Smoothing allows it to achieve better certification results than with previously considered non-SOTA models. We compare the results with the BEiT-B results reported in the paper.
>
> We do not provide the results for Segformer because its code comes with a license that prohibits (among others) usage in a commercial R&D lab.
>
> Table A. Certified recovery on ADE20K validation set (2000 images). mIoU - mean intersection over union, mR - mean recall, mR big - mean recall for "big" classes, cmR - certified mean recall, cmR big - certified mean recall for "big" classes. %C - mean percentage of certified and correct pixels in the image.
>
> | mode		    | patch size	|mask	    | mIoU      | mR big     | cmR big     | mR all     | cmR all     | %C        |
> |:--------------|:--------------|:----------|:----------|:-----------|:------------|:-----------|:------------|:----------|
> | BEiT-L    | 0.5%		    | column	| **28.64** | **71.95**  | **50.84**   | **34.65**  | **16.04**   | **47.76** |
> | 		        | 		        | row		| 18.82 | 53.77  | 21.24   | 22.74  | 5.95   | 32.30 |
> | 		        |       		| 3-mask	| 22.40 | 64.83  | 33.96   | 26.89  | 8.97   | 39.59 |
> |   		    | 		        | 4-mask	| 19.93 | 60.90  | 25.03   | 24.22  | 6.43   | 35.01 |
> |
> | BEiT-B | 0.5%		    | column	| **24.92** | **60.77**  | **41.26**   | **29.84**  | **12.98**   | **46.22** |
> | 		        | 		        | row		| 16.33 | 46.91  | 16.72   | 19.51  | 4.83   | 31.71 |
> | 		        |       	    | 3-mask	| 19.90 | 56.90  | 26.51   | 23.86  | 7.54   | 38.64 |
> |   	   	    | 		        | 4-mask	| 18.82 | 52.96  | 23.75   | 22.56  | 5.87   | 34.36 |
>
> Table B. Certified detection on ADE20K validation set (2000 images). mIoU - mean intersection over union, mR - mean recall, mR big - mean recall for "big" classes, cmR - certified mean recall, cmR big - certified mean recall for "big" classes. %C - mean percentage of certified and correct pixels in the image. FAR - mean false alert rate (lower is better).
>
> | model		    | patch size	| mask		    | mIoU (clean) | mR big (clean)     | cmR big     | mR all (clean)     | cmR all     | %C        | FAR $\downarrow$ |
> |:--------------|:--------------|:--------------|:-------------|:-------------------|:------------|:-------------------|:------------|:----------|:----------|
> | BEiT-L     | 1.0%		    | column	    | 56.33 | 74.26  | **61.15**   | 68.40  | **35.88**   | **65.44** | **19.51** |
> | 		        |       		| row   		| 56.33 | 74.26  | 52.77   | 68.40  | 30.25   | 60.48 | 24.47 |
> |
> | BEiT-B | 1.0%		    | column	    | 53.08 | 70.92  | **57.33**   | 64.45  | **32.55**   | **63.55** | **20.04**  |
> | 		        |       		| row   		| 53.08 | 70.92  | 50.05   | 64.45  | 26.65   | 58.34 | 25.24 |
>
>
> [1] Hangbo Bao, Li Dong, Songhao Piao, and Furu Wei. BEit: BERT pre-training of image transformers. In ICLR 2022.
>
> 2. "Hope the author can provide the code to check the reproducibility of this paper."
>
> We plan to publicly release the code upon acceptance, subject to the ongoing code release process.

---

> > ### Comment · Reviewer_XUU4 · 2022-12-01
> > **Thanks**
> >
> > Thanks for your rebuttal and additional results. I am not very familiar with the license policy. Just from the impact side, I think it would be more impact if we can use the STOA backbone. Nonetheless,  I tend to accept this paper.

---

### Author Response · Authors · 2022-11-17
**Rebuttal Revision**

We thank all the reviewers for their time and effort. We have made the following modifications in the rebuttal revision of the paper:

-    added the ablation studies with respect to the image inpainter in Table 9
-    added the experimental results for the BEiT-L segmentation model in Tables 6 and 7
-    added the limitations discussion in Section 5.4
-    made the proofs of Theorems 1, 2 in the Appendix A more detailed, added schematic illustrations for the proofs in Figures 7, 8
-    updated Figure 3 with more suitable illustrations
-    fixed a small typo in the caption of Figure 2

---

### Decision · Program_Chairs · 2023-01-20

**Decision:**

Accept: poster

**Justification For Why Not Higher Score:**

The proposed Demasked Smoothing certification requires an upper bound on the size of the expected patch. This is a limiation in this work but seems to be acceptable so far.

**Justification For Why Not Lower Score:**

This paper is one of the first certified defenses proposed for semantic segmentation.

**Metareview: Summary, Strengths And Weaknesses:**

In this paper, the authors present a Certified Defences method, Demasked Smoothing, Against Adversarial Patch Attacks on the Semantic Segmentation problem. The authors claim that in Demasked Smoothing, any segmentation model can be applied without particular training, fine-tuning, or restriction of the architecture, and both for certified detection and certified recovery can be achieved using different masking strategies.

The reviewers appreciate that (1) this paper is well-organized and is easy to follow and (2) this paper is one of the first certified defenses proposed for semantic segmentation.
The weaknesses raised by the reivewers have been properly addressed, including experimental evaluation, comparisons with previous works, and clarification of certified defense,

Given the above reasons and the reviewers consistently give positive comments, the AC recommends to accept this paper.


**Note From Pc:**

if the above contains the word "oral" or "spotlight" please see: "oral" presentation means -> notable-top-5% and "spotlight" means -> notable-top-25%. As stated in our emails, we are disassociating presentation type from AC recommendations